METHODS

# Identifying promoter sequence architectures via a chunking-based algorithm using non-negative matrix factorisation

**Sarvesh Nikumbh**[1,2]*, **Boris Lenhard**[1,2]*

**1** Computational Regulatory Genomics, MRC London Institute of Medical Sciences, London, United Kingdom, **2** Institute of Clinical Sciences, Faculty of Medicine, Imperial College London, Hammersmith Hospital Campus, London, United Kingdom

* s.nikumbh@imperial.ac.uk (SN); b.lenhard@imperial.ac.uk (BL)

## Abstract

Core promoters are stretches of DNA at the beginning of genes that contain information that facilitates the binding of transcription initiation complexes. Different functional subsets of genes have core promoters with distinct architectures and characteristic motifs. Some of these motifs inform the selection of transcription start sites (TSS). By discovering motifs with fixed distances from known TSS positions, we could in principle classify promoters into different functional groups. Due to the variability and overlap of architectures, promoter classification is a difficult task that requires new approaches. In this study, we present a new method based on non-negative matrix factorisation (NMF) and the associated software called seqArchR that clusters promoter sequences based on their motifs at near-fixed distances from a reference point, such as TSS. When combined with experimental data from CAGE, seqArchR can efficiently identify TSS-directing motifs, including known ones like TATA, DPE, and nucleosome positioning signal, as well as novel lineage-specific motifs and the function of genes associated with them. By using seqArchR on developmental time courses, we reveal how relative use of promoter architectures changes over time with stage-specific expression. seqArchR is a powerful tool for initial genome-wide classification and functional characterisation of promoters. Its use cases are more general: it can also be used to discover any motifs at near-fixed distances from a reference point, even if they are present in only a small subset of sequences.

## Author summary

Transcription of genes by RNA polymerase II enzyme is known to begin at specific positions in parts of DNA sequence called promoters. These positions, called transcription start sites, are chosen by protein complexes binding to sequence signals in the nearby DNA, either upstream or downstream of them. These protein complexes then help recruit RNA polymerase II at a specific location from which it will choose a specific transcription start site. The set of sequence signals that governs these events at each promoter constitutes its promoter sequence architecture. Different organisms show diversity in promoter

**Data Availability Statement:** All data for reproducing results in this manuscript is deposited at Zenodo with DOI 10.5281/zenodo.7692742 Software is available at: http://www.bioconductor.org/packages/seqArchR Code to reproduce

manuscript is available at: https://github.com/snikumbh/reproducible-seqArchR-manuscript.

**Funding:** Acknowledge funding from Wellcome Trust (Joint-Investigator award 106955/Z/15/Z) to B.L., and core funding from Medical Research Council (award MC_UP_1102/1) to BL (which provided salary for S.N. and running costs for both). The funders had no role in study design, data collection and analysis, decision to publish, or preparation of the manuscript.

**Competing interests:** The authors have declared that no competing interests exist.

sequence architectures. Even within an organism, different promoter architectures are characteristic of different kinds of genes such as those that are tissue-specific vs those that are ubiquitously expressed across tissues. In this paper, we present seqArchR, a method using non-negative matrix factorisation (NMF) for clustering of promoter sequences into their characteristic promoter sequence architectures *de novo*. We show that seqArchR is faster than state-of-the-art approaches and gives better or similar results on both simulated and real promoter sequences. We apply seqArchR on promoters across stages of embryonic development in fruit fly and zebrafish and show it reveals changes in relative use of promoter architectures over time. We also show that seqArchR can classify human promoter sequences, demonstrating its power to discover promoter architectures across very different genomes.

This is a *PLOS Computational Biology* Methods paper.

## Introduction

Promoter sequences are stretches of DNA flanking the transcription start site (TSS) of a gene. Historically, a window spanning −40 to + 40 base pairs (bp) (upstream denoted by '–' and downstream by '+') relative to the TSS is called the core promoter sequence. Further upstream from −40 to −250 bp is called the proximal promoter region [1]. These regions work together with enhancers—which can be anywhere in the genome, upstream or downstream of the gene they affect—to regulate gene expression. Transcription of genes by RNA polymerase II (RNA PolII) happens by assembly of the pre-initiation complex of RNA PolII and different basal transcription factors (TFs) [2]. These TFs recognise different sequence motifs present in the core promoters for binding.

However, there is no universal promoter architecture. Not all motif elements are present in all core promoters. For instance, TATA-box, which is bound by the TATA-box binding protein (TBP), is a popular textbook example of a core promoter motif, but at most ∼20% of the known core promoters of protein-coding genes in animals contain a TATA-box [2]. Furthermore, transcription initiation can be 'focused' or 'dispersed' [3, 4]. In focused transcription, there is a single nucleotide (nt) position where most of the transcription begins. In dispersed transcription there are multiple weaker start positions spread up to ∼100 bp within the promoter, and the dominant position is determined by the distance from the + 1 nucleosome [5, 6]. The resultant promoters are thus called *sharp* and *broad* respectively. Sharp promoters are overwhelmingly associated with tissue-specific genes, while broad promoters are associated with constitutively expressed genes and a subset of tissue restricted ones (reviewed in [7]). The most important realisation about different promoter architectures is that there is a TSS-selection determining sequence element at near-fixed distance from the dominant TSS (Fig 1). Different TSS selection mechanisms suggest that there are multiple fundamentally different promoter architectures, and that they are functionally specialised. Heterogeneity in architectures at the organism-level gives rise to additional diversity [3, 7, 8]. Also, the advent of chromatin interaction experiments in the last decade has enabled better exploration of the enhancer-promoter interaction specificity. Thus, studying and defining the promoter sequence architectures that give rise to such specificities in transcription regulation mechanisms is a key step in understanding transcriptional regulation.

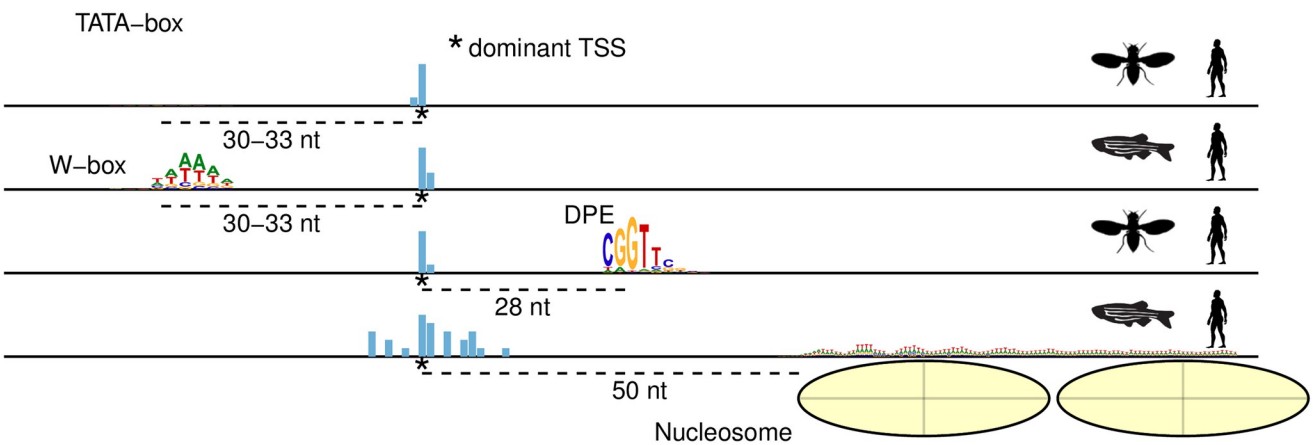

**Fig 1. Schema depicting how, across organisms, different promoter architectures have a TSS-selection determining sequence element at near-fixed distance from their dominant transcription start sites.** nt = Nucleotide(s). The silhouettes for Drosophila, Danio rerio and Homo sapiens were obtained from PhyloPic (http://phylopic.org). Licenses: "CC0 1.0 Universal Public Domain Dedication" for Drosophila (link); "CC0 1.0 Universal Public Domain Dedication" for Danio rerio (link); and "CC0 1.0/Universal Public Domain Dedication" for Homo sapiens (link).

Promoter sequence architectures are not always characterised just by occurrences of short sequence motifs but at times also by the sequence composition, for example, GC-richness or CpG island promoters. Using motif finding algorithms to process promoter sequences, as is traditional, has limitations. First, they cannot identify differences in architectures stemming from overall sequence compositions. Second, in scenarios where promoter sequences are indeed characterised by motifs, scanning the sequences for motifs from a database (when available) limits the study to looking at only known motifs. These known motifs often do not explain much of the diversity among the promoter sequences. In case of novel organisms for whom no databases of known motifs exist yet, a *de novo* motif search can be performed. But most motif finding algorithms expect the motifs to be statistically over-represented in the promoter sequences in comparison to background sequences. Such approaches can fail to identify motifs that are present in a relatively small subset of sequences and therefore not statistically 'significant', and do not provide precise position information of the motifs. Furthermore, they also often miss discovering low-complexity or degenerate motifs. New approaches, which can overcome the above limitations in using motif finding algorithms and identify architectures in promoter sequences *de novo* are required to study promoter sequence architectures.

Here, we present seqArchR, an unsupervised approach using non-negative matrix factorisation (NMF) [9] for *de novo* characterisation of diversity in promoter sequence architectures. seqArchR does not require background sequences. It needs no information on the kind (motifs- or sequence composition-based) or number of architectures expected to be present in the input sequences. When architectures characterised by motifs may be present, it does not require any specifications on the number of motifs, their sizes or any gaps in motifs or between them when they work in a cooperative fashion. seqArchR's chunking-based approach can reveal architectures harboured by very few sequences out of the whole input. This is often helpful to precisely tease out minor positional variations in features which can be functionally relevant. For instance, the spacing between the TATA-box and the TSS is known to correlate with tissue specificity [10].

seqArchR is inspired by 'No promoter left behind' (NPLB) [11], the current state-of-the-art method identifying such sequence architectures *de novo* from aligned promoter sequences.

NPLB uses a Bayesian approach to jointly learn features and important positions in an architecture. It identifies the number of architectures best explaining the complete set of promoter sequences by cross-validation where the model with the highest loglikelihood is chosen. NPLB has a relatively long execution time making it less effective as an exploratory tool. It takes about 10 hours to process ca. 6600 promoter sequences in Drosophila compared to seqArchR's 42 minutes when run serially to obtain comparable results. Additionally, seqArchR gives the user complete control over the granularity of identified clusters which is often useful in the exploratory setting.

Since Lee and Seung's work [9], NMF has become a popular technique finding wide applications across domains. Some earlier applications of NMF in biology include those for the inference of biological processes [12], sequence motifs characterisation [13], and multi-omics analysis [14]. In particular, Hutchins et al. [13] used NMF for characterizing sequence data sets by identifying position-dependent motifs in them. In their approach, the input matrix encodes the counts of the *k*-mers and their occurrence positions in the given collection of sequences. This approach assumes that all sequences in the data set contain the same motifs or minor variations thereof. Although this approach is well-suited for identifying protein-DNA binding sites from ChIP-seq like data, it assumes a single mode of binding and does not cater to the scenario of multiple modes of protein-DNA binding [15]. In comparison, seqArchR makes no such assumptions and models the collection of sequences to harbour any number of architectures, each with an unknown number of motifs– or non-motifs–based sequence features.

With seqArchR, we demonstrate the general applicability of NMF for *de novo* identification of near fixed-position motifs from sequences, and simultaneously clustering the sequences based on the identified motifs. We note that seqArchR is not a general sequence motif discovery tool. It will not detect those overrepresented promoter motifs that are found at a wide distribution of distances from transcription start sites, such as GC-box and CAAT-box in vertebrates, or DNA recognition element (DRE) in Drosophila. After seqArchR identifies architecturally distinct clusters based on TSS-associated sequence elements, the obtained clusters can be analysed using general motif finders to investigate which non-TSS determining motifs are associated with which architecture. (See Discussion section for further details.).

This paper describes seqArchR in complete detail and presents results from computational experiments demonstrating its efficacy. We compare seqArchR with NPLB in analysing a simulated dataset where the ground truth of clusters/architectures is known, and a set of CAGE-based core promoter sequences from Chen et al. [16] (S1 Text). Furthermore, we show results from processing CAGE-based promoter sequences in fruit fly [17], zebrafish [18] and human [19, 20] to demonstrate seqArchR's ability to seamlessly identify heterogeneous architectures across organisms with large differences in promoter sequence composition. seqArchR is provided as an R/Bioconductor package accessible at https://www.bioconductor.org/packages/seqArchR or https://snikumbh.github.io/seqArchR.

## Results

### seqArchR: A novel algorithm using non-negative matrix factorisation for *de novo* identification of sequence architectures

To classify promoters based on their fixed-position determinants, we developed, seqArchR, a chunking-based iterative algorithm using NMF for *de novo* identification of architectural elements. The input to seqArchR is a (0, 1)-matrix which is a one-hot encoded representation of dinucleotide profiles of a gapless alignment of DNA sequences. As shown in the schematic

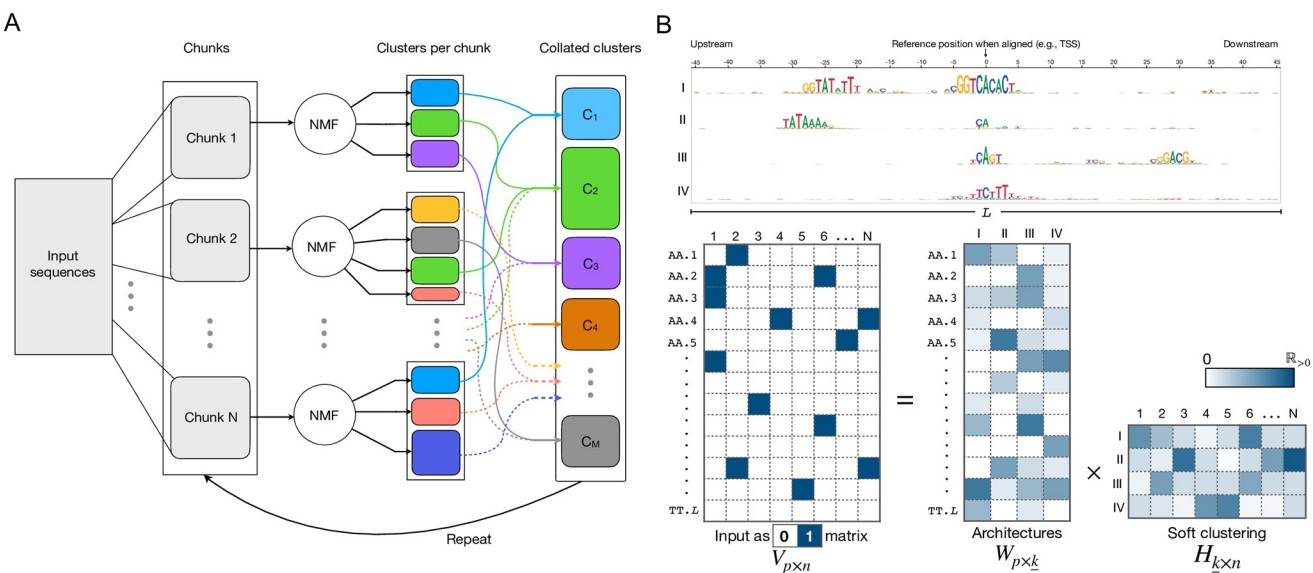

**Fig 2. Overview of seqArchR.** (**A**) Schematic showing seqArchR's chunking-based, iterative algorithm. (**B**) Schematic describing input to seqArchR and the factorisation output. For each chunk of sequences being processed with NMF, the sequences are represented as a one-hot encoded matrix (hence, 0/1 matrix), denoted in the schematic by matrix $V_{p \times n}$; matrices $W_{p \times k}$ and $H_{k \times n}$ are respectively the basis matrix and coefficients matrix obtained upon factorisation. $n$ denotes the number of input sequences, $p$, the number of features, $k$, the optimal number of dimensions selected for the low-rank representation and $L$, the length of the input sequences. The schematic depicts one-hot encoding of the dinucleotide profile of sequences. One can use the mono- or dinucleotide profile of sequences.

in Fig 2, seqArchR processes the whole collection of input sequences one chunk (subset of sequences) at a time. The (0, 1)-matrix for each chunk of sequences is processed with NMF. NMF decomposes the matrix into two low-rank matrices—the basis matrix and the coefficients matrix. Columns of the basis matrix (a.k.a. basis vectors) represent the different potential architectures, and along its rows are the loadings for the features per architecture (see Fig 2B). On the other hand, in the coefficients matrix, where each column corresponds to one sequence, its rows hold the per-sequence coefficients for each architecture. A sequence with a high coefficient for an architecture is expected to harbour features captured by that architecture. For example, consider matrix $H_{k \times n}$ in Fig 2B, the first and second sequence have the highest coefficient for architecture I among all four architectures, and are expected to harbour sequence features prominent for architecture I (see corresponding to the sequence logos on the top). Because the input matrix already encodes the position information of the nucleotides in the sequences, the factorisation identifies the characteristic architectures (combinations of nucleotides over-represented at specific positions) describing different subset of sequences.

Per chunk, seqArchR performs a model selection procedure to find the appropriate number of basis vectors suitable to represent the set of sequences in the chunk in a lower-dimensional space. Two model selection procedures are available: stability– and cross-validation–based (see the Methods section for details). This decomposition is used to obtain clusters of sequences characterised by a diverse set of sequence features.

After all chunks have been factorised and sequence clusters obtained, similar clusters from different chunks are collated. Similarity of clusters is judged based on the similarity of corresponding NMF basis vectors (see Methods section for additional details). The collated set of clusters may need further processing to weed out any mis-assignments or to crisply identify

architectures with shared motifs or those with minor positional shifts. In this case, seqArchR iterates over the collated clusters treating each of them as separate chunks and repeating the above steps. The number of iterations required depends on the structure in the input data. The algorithm is described in detail in the Methods section.

## seqArchR is fast and attains high accuracy on simulated data

In order to demonstrate seqArchR's efficacy, we performed computational experiments on simulated data as follows. We generated a set of 1000 simulated DNA sequences, each 100 nucleotides long. All nucleotides at any position in these sequences appeared with a uniform probability of 0.25. We planted different motifs at specific positions in these sequences such that they resulted in four clusters with as many architectures. These are described in Table 1.

All motifs except those in cluster A were planted with mutations, as follows. For planting a motif in a sequence, any one position in the motif was randomly selected for mutation. The nucleotide at this position was then mutated with probability (mutation rate) {0.1, 0.2, 0.3, 0.4, 0.5}. This resulted in five variations of DNA sequences—one mutated position × five mutation rates for all motifs. This procedure was repeated, selecting two and three positions for mutation per motif (instead of one), giving us a total of 15 variations. For each of these variations, we performed experiments with chunk sizes {200, 500, 1000}. Thus, all parameter combinations resulted in a total of 45 datasets to be processed by seqArchR. For robustness, each of these was repeated 10 times with different random seeds. Here, we used stability-based model selection.

For comparison, we also processed these 15 datasets with NPLB [11]. Performance results of seqArchR and NPLB are presented next. All reported timing information is for experiments performed in parallel on 16 cores of Intel(R) Xeon(R) Gold 6154 CPUs.

**Performance as adjusted Rand index (ARI).** Performance is reported using the adjusted Rand index (ARI) which measures the similarity between two clustering solutions or a clustering and ground truth (when available), and, is corrected for chance. As seen in Fig 3A, irrespective of the chunk size, seqArchR attains a high ARI value when the mutation rate and the number of mutated motif positions is low. For any given chunk size, the performance degrades with increase in the mutation rate, while for the same mutation rate, increasing the number of mutated motif positions hardly affects performance. For smaller chunk size, seqArchR's performance shows a slight increase in variance with increase in mutation rate. With a larger chunk size, seqArchR demonstrates very low variance in comparison to NPLB which shows high variance for all mutation rates.

Fig 3E and 3F show the time taken (in minutes) by seqArchR and NPLB. In Fig 3E, the colours denote time taken by seqArchR to complete different iterations. We observe that processing the whole data in many, small chunks, takes more time overall. This is expected: for each chunk, the stability-based model selection procedure starts from checking for just two clusters and continues checking further values until the bounding threshold on the instability of identified cluster features is satisfied (see more details of the algorithm in the Methods section).

**Table 1. Sequence motifs corresponding to four architectures/clusters in the simulated data.**

| Cluster | Motifs | Occurrence position | #Sequences |
|---|---|---|---|
| A | Dinucleotide repeat AT starting at 10th nt | every 10 nt | 200 |
| B | GATTACA and GAGAG | 40 and 60 | 350 |
| C | GAGAG | 60 | 150 |
| D | TCAT and GAGAG | 40 and 80 | 300 |

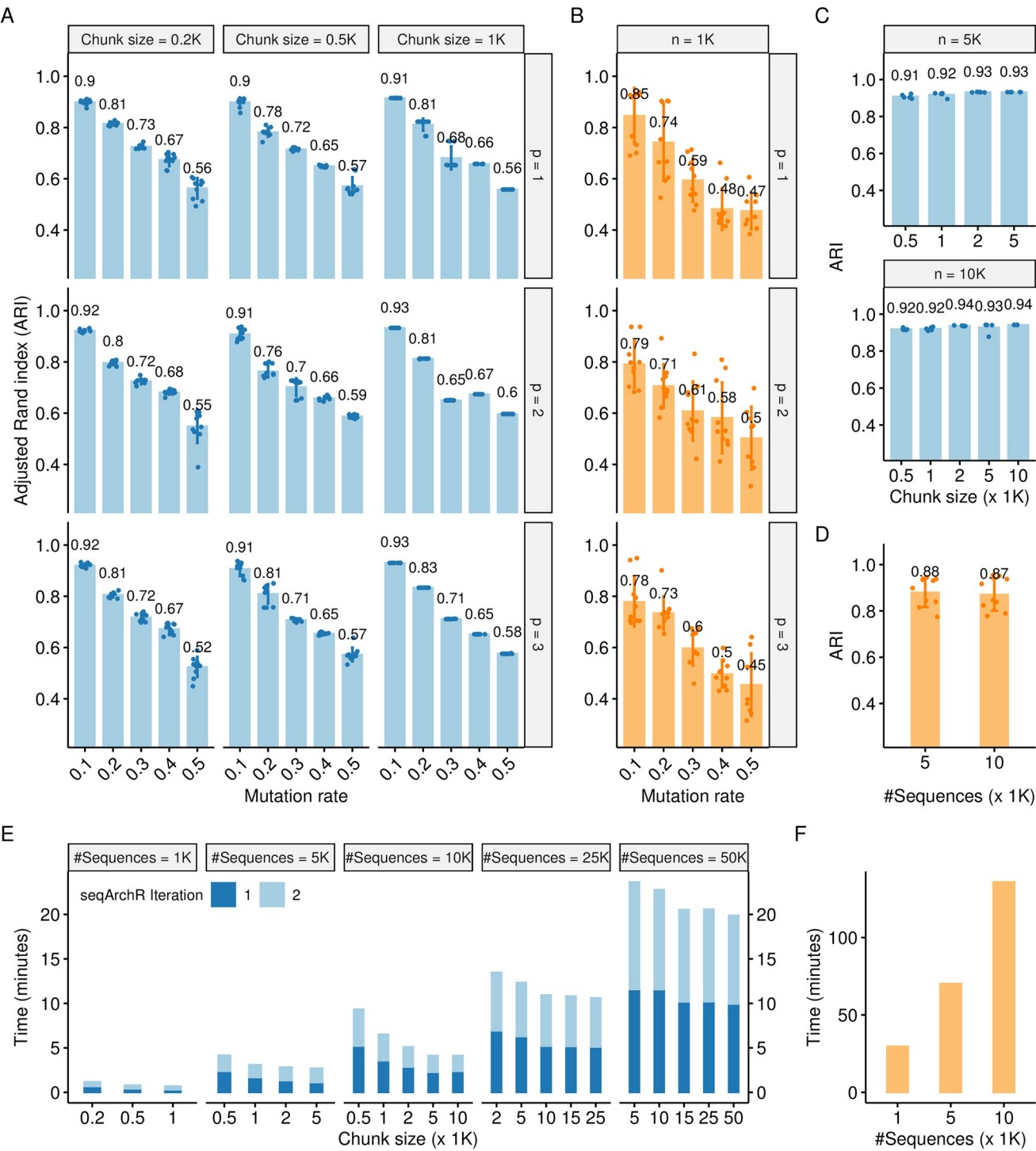

**Fig 3. Assessment of performance of seqArchR on simulated data.** (**A**) Adjusted Rand index (ARI) attained by seqArchR with various parameter combinations on simulated data (1000 sequences) (data described in Table 1). Bar heights represent average over ten runs. Experiments are performed for various chunk sizes, mutation rates (m) and number of mutated motif positions (p). (**B**) ARI attained by NPLB for 1000 sequences setting. (**C, D**) ARI attained by seqArchR and NPLB respectively for two scaled up versions of the simulated data: 5000 and 10000 sequences. See main text for more details. (**E, F**) Time taken by seqArchR and NPLB respectively to process simulated data (see main text for details).

Since sequences from each cluster are distributed uniformly randomly across the complete set of sequences, for each chunk, the optimum number of clusters is expected to be around the true number of clusters. Thus, there is some redundancy of checking the lower values for every chunk. This redundancy reduces for larger chunk sizes.

## seqArchR identifies fixed elements of promoter sequence architectures in different organisms

In general, when dealing with DNA sequences, different organisms are known to present different challenges for sequence analysis methods. For example, the high GC and especially CpG content in human genomic sequence is known to affect approaches using motif over-representation [21, 22]. More specifically, with respect to gene promoters, there is diversity of sequence elements within promoters of one organism and also between those of different organisms. Different promoter types have different sequence composition, some of which are highly constrained, and some are not.

Drosophila promoter sequences have multitude of well defined sequence motifs viz. the TATA-box and various Ohler motifs [23, 24]. In comparison, promoters of vertebrates have relatively fewer well defined motifs; they are also characterised by the stretches of varying sequence compositions, e.g., CG/GC dinucleotide content *vs* A/T-rich W-box motifs, owing to different architectures (sometimes even intertwined) determining TSS selection at play during different stages of development [5, 25]. This diversity of promoter sequence elements reflects the diversity in mechanisms of transcription regulation. Thus, computational methods that work for one organism may not work off-the-shelf for other organisms. In the following sections, we evaluate seqArchR's ability to identify fixed-element promoter sequence architectures in fruit fly, zebrafish and humans. Note that promoter architecture clusters reported here for all three organisms are obtained by manually curating (post-processing) the raw clusters identified by seqArchR in its final iteration. This is discussed further in the "Curation of collated clusters" subsection in the Methods section.

### Fruit fly (*Drosophila melanogaster*)

Schor et al. [17] produced high resolution CAGE data and studied the effect of natural genetic variation on transcription strength and distribution of transcription start sites in gene promoters at different developmental stages in Drosophila. From the various samples available, we randomly selected one sample 'RAL28' (this number identifies the D. melanogaster line from Drosophila melanogaster Genetic Reference Panel (DGRP) that were used by Schor et al. [17]) to help demonstrate seqArchR's efficacy.

Specifically, we used seqArchR to identify promoter architectures at three stages of transition during embryogenesis—2–4 hours (h), 6–8h, and 10–12h after egg laying (AEL). Each stage was analysed with seqArchR separately. We used 91 bp-long sequences—45 bp upstream and downstream of the TSS—as core promoter sequences. Clustering results for D. melanogaster are reported from experiments using stability bound $10^{-8}$ (see Methods). Figs 4, 5, and 6 show the sequence logos of architectures of all clusters identified for the three respective transitions. For brevity, we use the following shorthand notation to refer to clusters at any transition: CnX or CnY or CnZ. Here, 'C' stands for cluster, n, the cluster number, and letters X, Y and Z denote the three transitions 2–4h, 6–8h, and 10–12h AEL. A single cluster, say cluster 3 at 2–4h, is named C3X, and multiple clusters, say clusters 2–6 or clusters 3 and 5 at 10–12h are named C2–6Z or C3,5Z respectively. Unless specified otherwise, all clusters are arranged in ascending order of their median interquantile width (IQW).

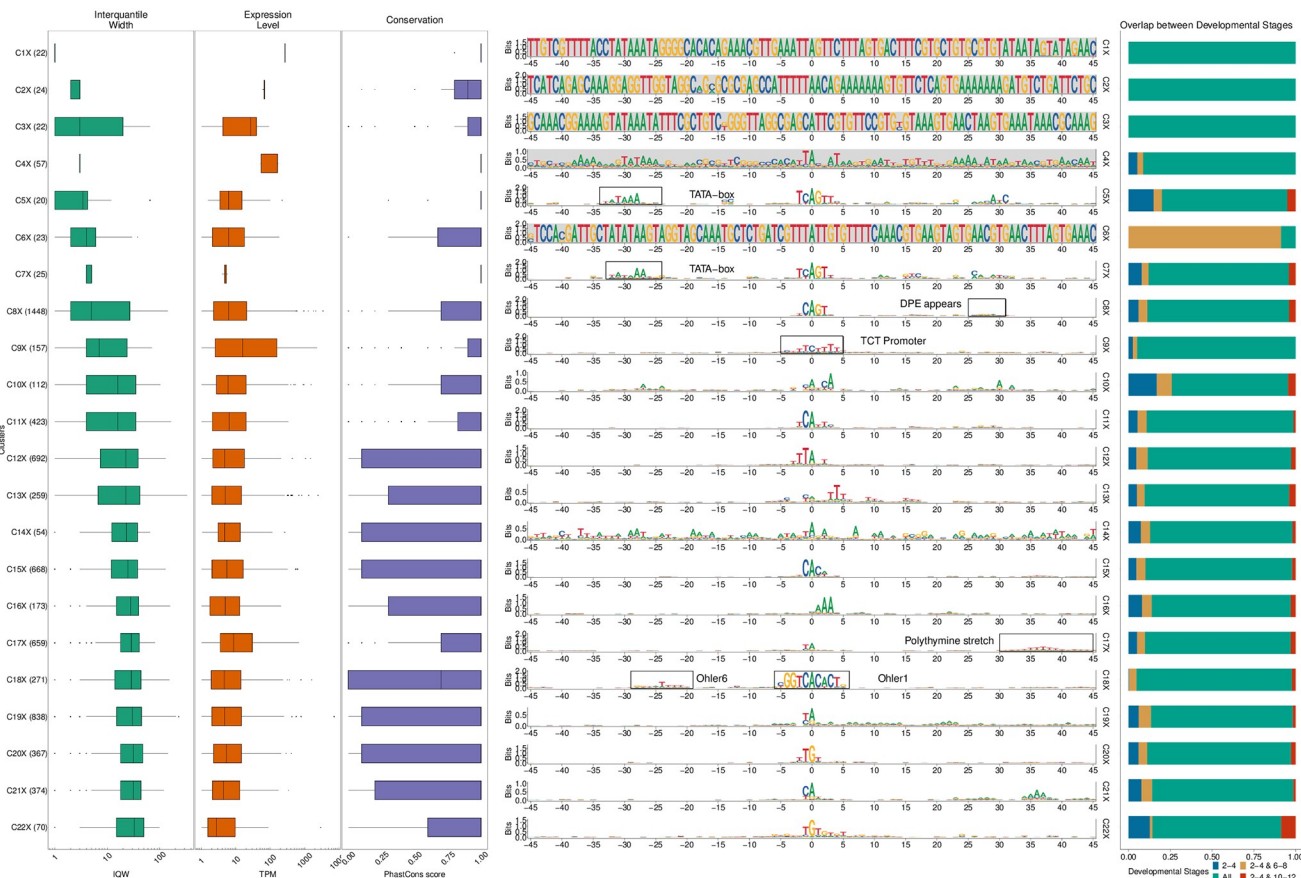

**Fig 4. Clusters and architectures identified by seqArchR for *D. melanogaster*, 2–4h AEL.** Sequence clusters arranged by the median interquantile widths (IQW) of CAGE TCs in seqArchR clusters (shortest on top, broadest at the bottom). From left to right: Box and whisker plots of per-cluster IQWs, TPMs, and PhastCons scores followed by per-cluster sequence logos, and stacked barplots showing proportion of TCs unique/shared between transitions. Sequence logos for histone gene clusters are shown with a grey background. 'All' denoting common between all stages. TPM, Tags per million.

**Motifs.** seqArchR identifies several of the known core promoter elements in Drosophila. These are directional motifs with a precise or variable positioning within promoters—abbreviated as DMp and DMv respectively. The identified DMp motifs include the `TATA`-box, initiator (Inr, canonical and other non-canonical ones), the downstream promoter element (DPE). Among DMv motifs, seqArchR identifies Ohler motifs 1 and 6, and another motif named DMv5 [23, 26]. seqArchR also identifies some of the non-directional motifs (NDMs) reported by FitzGerald et al. [26] such as the downstream regulatory element (DRE) or Ohler motif 2 (NDM4). The non-directional sequence elements are visible in the sequence logos of the raw clusters from seqArchR (see Figs F, G, and H in S1 Text). These were manually combined to obtain the final clusters as seen in the main figures (see Methods section, "Curation of collated clusters" subsection; and Figs F, G, and H in S1 Text).

`TATA` and DPE sequence elements are known to pair mutually exclusively with the Inr for architectures characteristic of tissue-specific and developmental genes respectively [27]. DPE appears very weak in the sequence logo for C8X (Fig 4), C7,10Y, (Fig 5) and C3,5Z (Fig 6) as against C1Y (Fig 5) due to collation of clusters with similar but stronger/weaker DPE signals resulting in an overall weaker DPE signal in the final cluster (Table 2). C8X with Inr+DPE architecture and C1Y with Inr+DPE1 (variant of DPE) architecture have a very short median

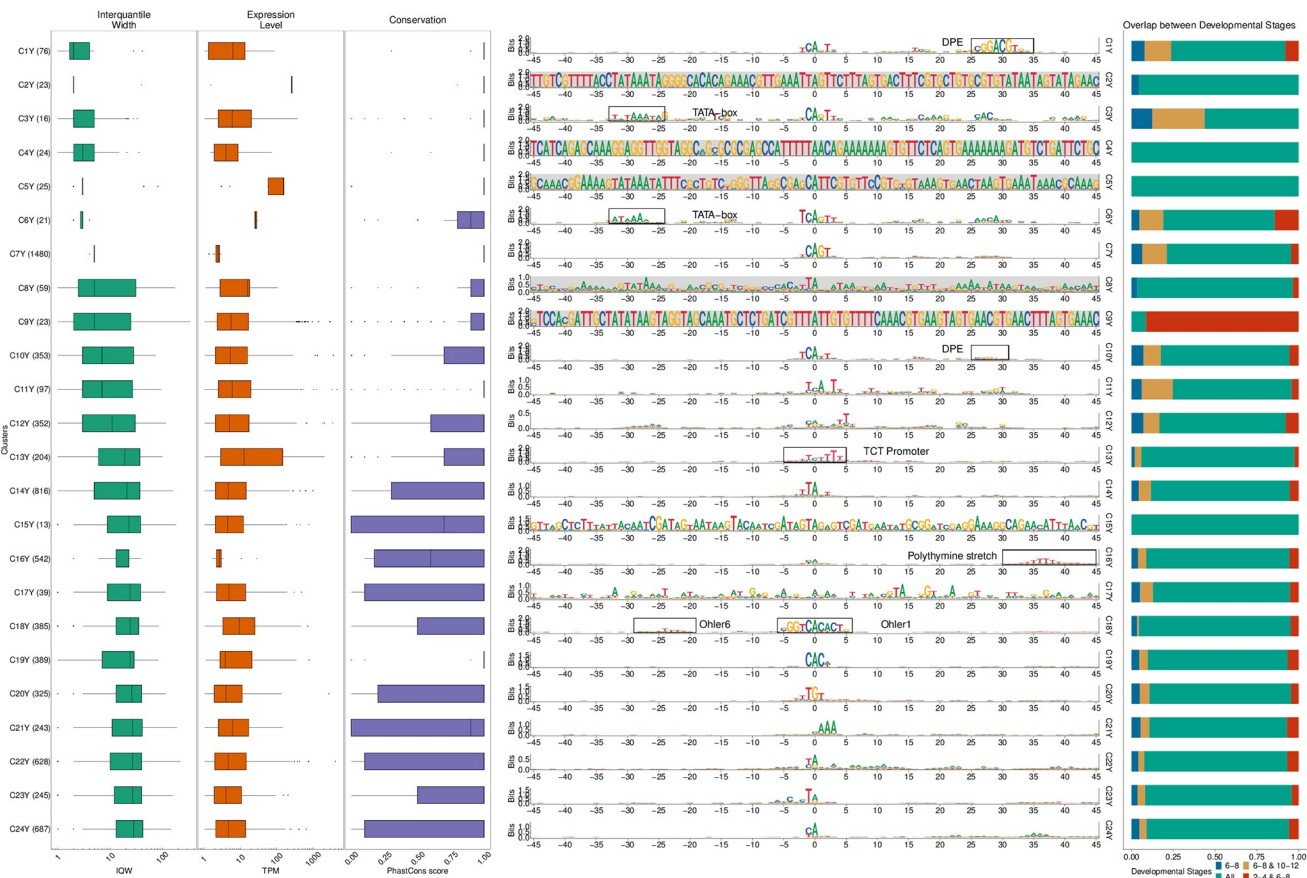

**Fig 5. Clusters and architectures identified by seqArchR for *D. melanogaster*, 6–8h AEL.** Sequence clusters arranged by the median interquantile widths (IQW) of CAGE TCs in seqArchR clusters (shortest on top, broadest at the bottom). From left to right: Box and whisker plots of per-cluster IQWs, TPMs, and PhastCons scores followed by per-cluster sequence logos, and stacked barplots showing proportion of TCs unique/shared between transitions. Sequence logos for histone gene clusters are shown with a grey background. 'All' denoting common between all stages. TPM, Tags per million.

IQW denoting highly focused transcription. C8Y combines the cognate initiator, Inr1, with the DPE.

With a strict stability bound $10^{-8}$, seqArchR identified a strong DPE signal at all three stages AEL, while a strong TATA-box architecture was only identified at 10–12h AEL (see Figs K, L and M in S1 Text). Some TATA-box promoters seemed to be misclassified. So, we processed the promoters at all three stages individually using more lenient bound values (see S1 Text, subsection "Ensuring identification of TATA-box at all stages in D. melanogaster development" for more details) to tease out any missed TATA signal. The final set of clusters reported here accommodate these previously missed TATA architecture clusters identified with lenient bound values.

TATA-box is observed at all three stages (Table 2). Of these, cluster C4Z, with, comparatively, the highest number of promoters, is the one where the TATA-box appears most downstream, nearest to the dominant TSS and combines with a canonical dinucleotide initiator $C_{-1} A_{+1}$, while the others combine with a longer, full Inr (consensus $TCA_{+1} KTY$) [23, 26].

Fig 7A compares the top-5 gene ontology (GO) terms that the TATA-box and DPE clusters are enriched for. Interestingly, many GO terms enriched in the DPE clusters are common across stages, while all the TATA-box clusters are enriched for mostly different GO terms, but

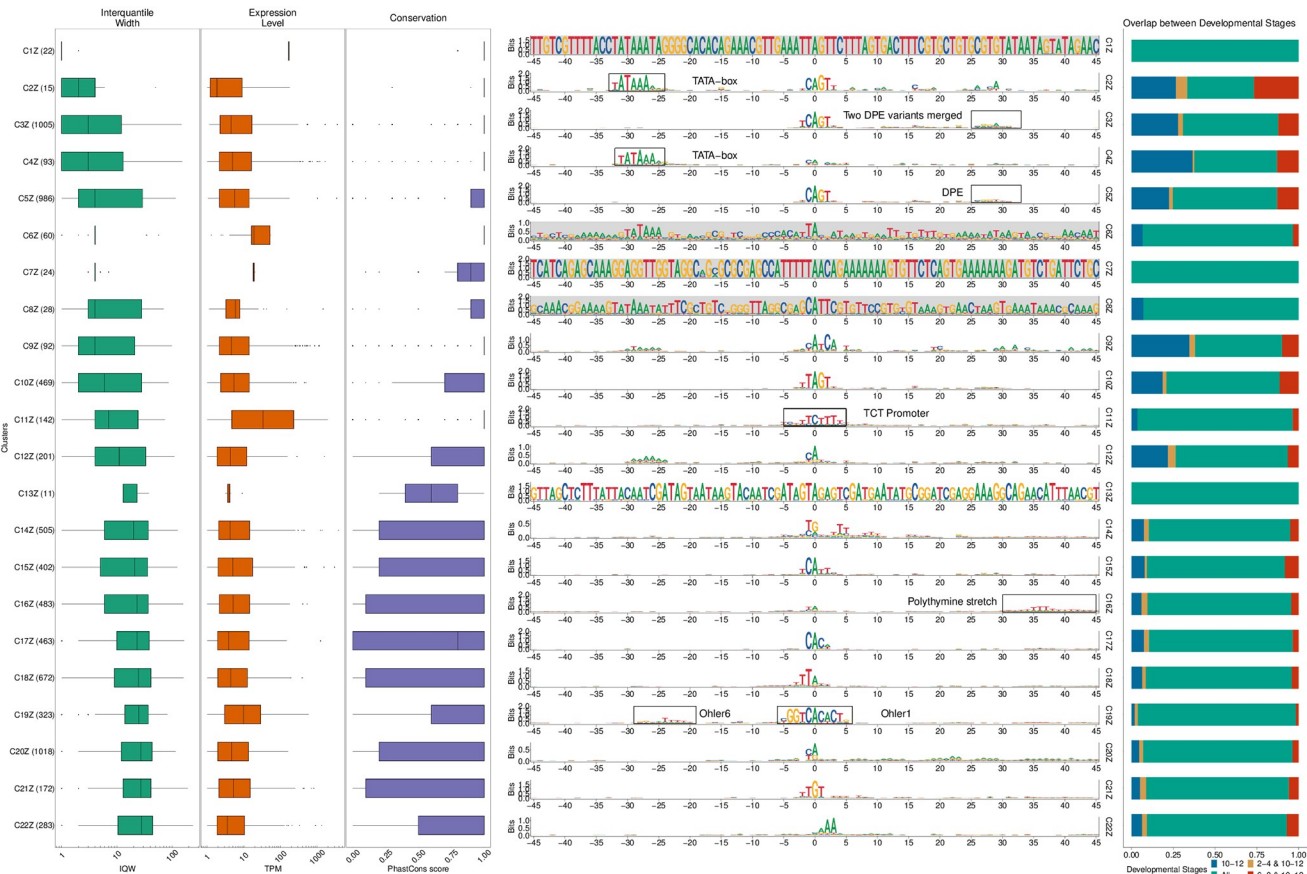

**Fig 6. Clusters and architectures identified by seqArchR for *D. melanogaster*, 10–12h AEL.** Sequence clusters arranged by the median interquartile widths (IQW) of CAGE TCs in seqArchR clusters (shortest on top, broadest at the bottom). From left to right: Box and whisker plots of per-cluster IQWs, TPMs, and PhastCons scores followed by per-cluster sequence logos, and stacked barplots showing proportion of TCs unique/shared between transitions. Sequence logos for histone gene clusters are shown with a grey background. 'All' denoting common between all stages. TPM, Tags per million.

all related to development, e.g., eye morphogenesis, central nervous system development, extracellular matrix, cuticle development, head development, and protein (re-)folding (see Fig X in S1 Text for top-10 enriched GO terms). Among the two minor early TATA clusters seen at 6–8h AEL, C3Y is dominated by heat shock proteins, and only a small number of tissue-constrained promoters appear in C6Y that seems to be specifically involved in expression in the eye (Fig 7A).

**Table 2. Clusters with DPE motifs at all three developmental stages in D. melanogaster.**

| Clusters | Stage | Raw clusters collated | Supplementary Figure |
|---|---|---|---|
| C8X | 2–4h AEL | 27, 30, 31, 32, 33 | Fig F in S1 Text |
| C1Y | 6–8h AEL | 3 | Fig G in S1 Text |
| C7Y | 6–8h AEL | 1, 2, 5, 7, 9 | Fig G in S1 Text |
| C10Y | 6–8h AEL | 4, 6, 46, 47 | Fig G in S1 Text |
| C3Z | 10–12h AEL | 2, 4, 5, 10 | Fig H in S1 Text |
| C5Z | 10–12h AEL | 1, 3, 7, 8, 9 | Fig H in S1 Text |

Comparison of GO terms enriched for different promoter architectures in D. melanogaster across developmental stages

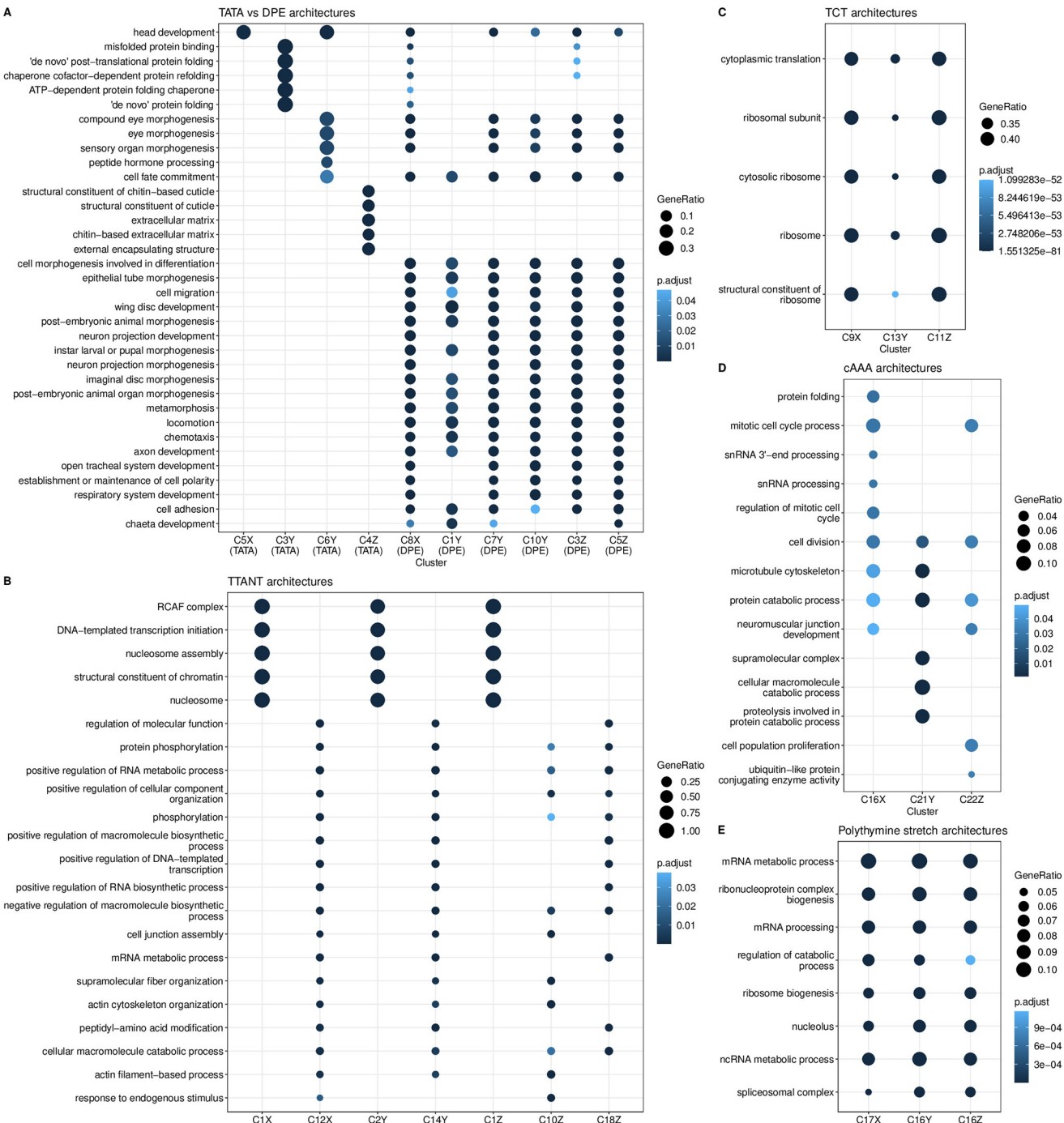

**Fig 7. Visualisation of GO terms enriched for various clusters at different developmental stages of Drosophila melanogaster.** Top-5 enriched GO terms are shown for clusters with: (**A**) `TATA` vs `DPE` architectures; (**B**) `TTANT` architecture; (**C**) `TCT` architecture; (**D**) `cAAA` architecture; and (**E**) polythymine stretch architecture.

In C9Z, there are two initiators, the canonical CA and longer, full Inr (consensus $TCA_{+1}$ KTY) [23, 26], their $A_{+1}$ 3nt apart. We speculate that the canonical CA is paired with the TATA-box at −31 bp from the TSS, and the full Inr 3 bp downstream is paired with a weaker DPE which is also correspondingly shifted by 3 bp downstream. C12Z has an Inr with a TATA-like sequence at −30. Many CTSSs in these tissue-specific architectures (with TATA and DPE) are unique to 10–12h (relatively higher percentages of sites unique to a transition point) denoting that these start sites are not active earlier.

TGT motif at initiator position is found in all timepoints—C20,22X and C20Y and C21Z. TTAGT is identified at the initiator location in C10Z. A variant of this initiator with a weaker G is found in C12X, C14Y. C10Z has a few different GO terms enriched in comparison to C12X and C14Y, both of which have many common enriched terms as seen in Fig 7B. See Fig X in S1 Text for the top-10 enriched GO terms.

**Functional specialisation of clusters.** As expected, TATA-box and DPE are among the top-ranked, focused architectures (clusters with lower median IQW) overall. The polypyrimidine initiator, TCT architecture, is observed for genes responsible for translational machinery, such as the ribosomal protein genes (see top-5 enriched GO terms in Fig 7C). These genes are known to have ubiquitously high expression [28]. The corresponding clusters across stages, namely C9X, C13Y and C11Z, have the highest median TPM values while still being relatively sharp.

**Detection of recently expanded gene families.** seqArchR can identify promoters associated with histone genes which are known to have sequence repeats. These are identified across all timepoints. Specifically, C1–4,6X, C2,4,5Y and C8–9Y, and C1Z and C6–8Z are enriched for histone genes His1, His2A/2B and His4/4r and His3. This explains the very high information content of the sequence logo throughout the core promoter region. All CTSSs of His1 gene, namely C2X, C4Y and C7Z, have TSSs in 5' UTRs. Clusters C4X, C8Y, and C6Z, all clusters of His2B gene CTSSs, are contaminated with some non-histone ones (see section 4.2 in the S1 Text for more details on this contamination). We recommend that studies looking to generate a compendium of genome-wide CAGE-derived promoters and their properties for any non-model, as yet un(der)-studied, organism should report them but may choose to sequester them from any downstream analyses especially when these CTSSs are originating from non-promoter regions, e.g., 5' UTRs (Fig J in S1 Text).

**Novel Observations.** seqArchR also detects a novel positional motif consisting of a poly-thymine stretch, reminiscent of T-blocks (with >= 3Ts) common in C. elegans core promoters [29]. We observe such T-blocks at transitions 2–4h and 10–12h AEL. Specifically, C13X and C14Z, both contain T-blocks flanking the CTSS of core promoters in this cluster. T-stretches have been reported in human genome, too, where they were shown to be able to bind TBP [30].

C17X, C16Y and C16Z, all have A/T enrichment at 30 to 45 bp downstream of the CTSS, with a marked increase precisely at 35 bp. The GO terms enriched for these clusters are also similar—mainly including splicing- and catabolic/metabolic process-related terms (Fig 7E and Fig J in S1 Text). We speculate that this architecture is related to the reported pausing of the RNA PolII between + 20 bp and + 50 bp with the centre at + 35 bp from the TSS [31]. This arrangement places the front end of PolII just 10 bp away from the + 1 nucleosome border and enables their contact. Kwak et al. [32] also report on promoter-proximal PolII pausing (termed focused-proximal pausing) at 35–40 bp downstream of the TSS. Indeed, the gene RpS7, an example proximal-pausing gene from Kwak et al. [32] is present in these clusters.

**Temporal successions of motif use in Drosophila development.** A look at the different motifs present (or absent) in the promoter architectures at successive developmental stages, termed 'successions of motifs' here, can help gain insight into biologically meaningful

preferences across stages. When comparing promoter sequence architectures prevalent at different stages of Drosophila development, some trends immediately emerge. While the DPE is observed from 2–4h and beyond in large numbers (ca. 1500/25% of all promoters at each stage), very few promoters (only ca. 40) have a canonical TATA-box. This number grows beyond 100 only at 10–12h stage (C2,4Z), albeit proportionally still very low in comparison to the DPE clusters. This confirms the idea that the DPE is associated with developmental regulation, and is on average used earlier than TATA, which drives mostly tissue-specific expression in differentiated cell types [33]. Unlike TATA, DPE, TTT and initiator, other motifs identified by over-representation (e.g., Ohler motif 1/6, DRE, etc.) from Ohler et al. [23]) and FitzGerald et al. [26] do not occur at fixed distance from the TSS.

For completeness, we also analysed Drosophila melanogaster promoter sequences from modENCODE [16] with seqArchR to compare its performance on real promoter sequences with that of the current state-of-the-art approach, NPLB [11]. seqArchR identifies all architectures including variations of the TATA-box architectures with their precise positional changes. While seqArchR was run for 5 iterations for this data, it identifies a comparable set of clusters/architectures after three iterations and just 38 minutes compared to the 600 minutes taken by NPLB (for its first pass). See additional details in S1 Text, section 2.

## Zebrafish (*Danio rerio*)

Nepal et al. [18] produced the first single nucleotide resolution transcriptome (CAGE) data for a vertebrate, zebrafish, across different stages of embryonic development. We used seqArchR to identify different promoter architectures in zebrafish at three different developmental stages, from early to late, namely 64 cells (early maternal), 30% Epiboly/Dome (late maternal/early zygotic transition), and the Prim-6 stage (late zygotic). Promoter sequences at each stage were independently analysed with seqArchR to identify clusters and architectures. We use the same naming scheme, as used for D. melanogaster, for referring to clusters identified in zebrafish, except X, Y and Z here denote the three developmental stages of zebrafish analysed in this study.

Figs 8, 9 and 10 show the clusters identified for 64 cells stage, 30% Epiboly, and Prim-6 stage respectively.

**Motifs.** In comparison to Drosophila, core promoters of zebrafish (and other vertebrates) do not possess as much variety of sequence motifs at fixed distance from the dominant TSS. In zebrafish, across developmental stages, genome-wide transcription initiation occurs at $Y_{-1} R_{+1}$ (see Figs 8, 9 and 10; [18]). Motif $GGG_{+1}$ is seen in some clusters (C4X, C7Y, C7Z) where majority of the CTSSs are located in non-promoter regions, especially 3' UTR.

At 64 cells stage, which is only 2 hours post fertilisation (hpf), all architectures contain the canonical pyrimidine-purine initiator with an upstream TATA-box or a W-box—a TATA-like element, but more degenerate than canonical TATA box. This is the architecture used for maternal transcription in the oocyte [5]. The TATA-box containing cluster with the sharpest architecture among all is enriched for lectins (e.g., rhamnose-binding lectins) which are carbohydrate binding proteins (Fig 11B) and are known to provide essential innate immune response in early embryos in marine animals. While it is known that the W-box architecture is sufficient for oocyte expression, rhamnose-binding lectins are also known to be expressed in other types of cells in ovaries and also in gut epithelium [34], this might explain why they have a canonical TATA-box.

While clusters C6X at 64 cells and C13Y at 30% Epiboly/Dome stage have enrichment of WW downstream starting at ca. 50 bp, this signal is not periodic. In comparison, cluster C12Z, among the broader ones at the Prim-6 stage, exhibits a 10 bp periodicity in dinucleotide WW.

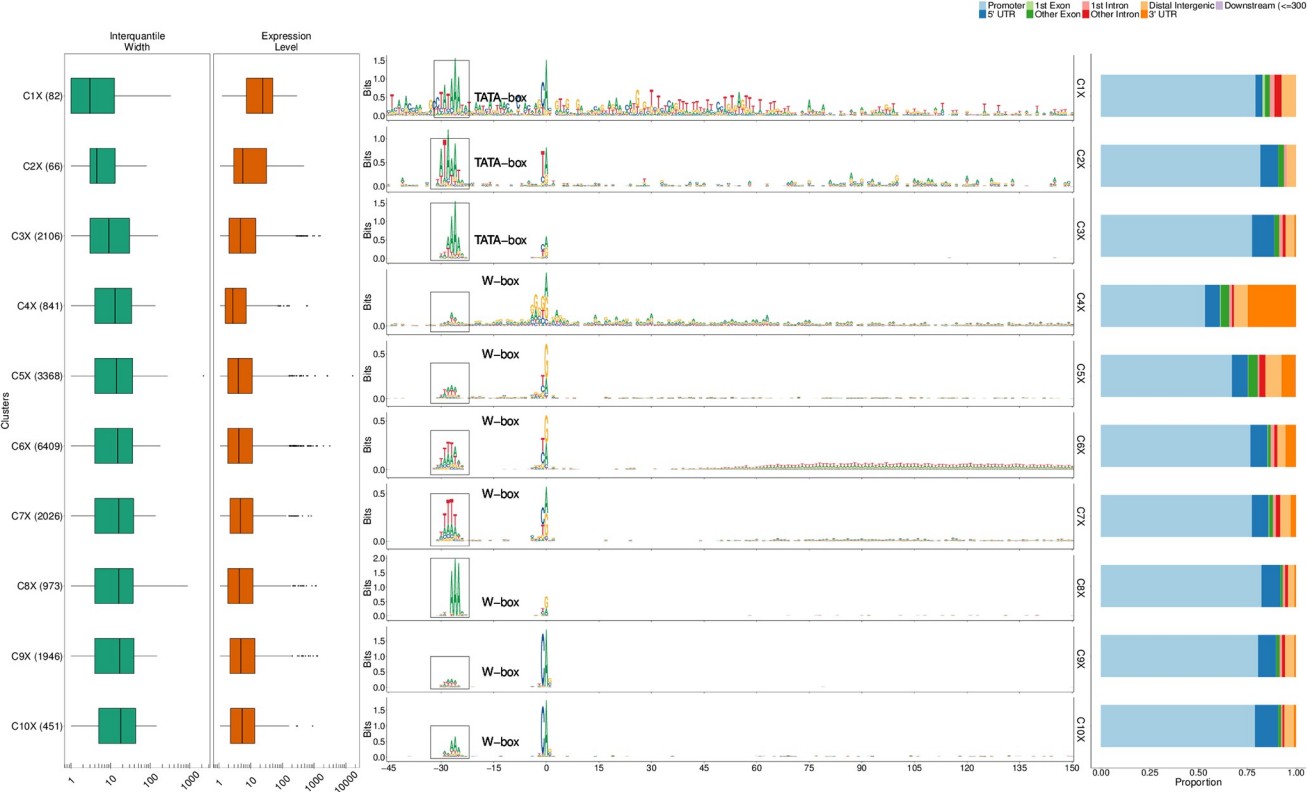

**Fig 8. Clusters and architectures identified by seqArchR for 64 cells stage.** Sequence clusters arranged by the median interquantile widths (IQW) of CAGE TCs in seqArchR clusters (shortest on top, broadest at the bottom). Each panel, from left to right: Box and whisker plots of per-cluster IQWs, and TPMs followed by per-cluster sequence logos, and stacked barplots showing proportion of different genomic annotations. TPM, Tags per million.

This periodicity is known to be an intrinsic property of promoters with a well-positioned + 1 nucleosome downstream of the nucleosome free region (NFR) spanning the promoter [6, 35]. Clusters C13,15Z also show a weak periodic signal visible in the WW motif occurrence heatmap (Fig AD in S1 Text). This is characteristic of broad promoter architectures where the position of the downstream nucleosome defines a so called 'catchment' area for transcription initiation [7].

**Functional specialisation of clusters.** Even though more than 90% of the stable RNA in 30% Epiboly/Dome stage is still maternally inherited [5], seqArchR already detects zygotically activated promoters and their specific architectures. Notably, it detects the promoters of the first wave of zygotic genome activation (ZGA), primarily coming from a dedicated gene cluster on chromosome 4 (clusters C2,9,12,14,15Y) (Fig 11A, and Fig AF in S1 Text). These clusters have genes encoding mir-430, which are present in hundreds of moderately divergent copies [36], and whose miRNA product specifically targets maternal RNAs for degradation [37, 38]. Additionally, at the Prim-6 stage, cluster C6Z contains predominantly forward strand promoters on located on chromosome 22 (Fig AG in S1 Text). The genes in this locus all belong to a family of NACHT-domain and leucine-rich-repeat-containing (NLR) proteins, involved in innate immunity. In zebrafish lineage, this family has been tandemly expanded into approximately 400 copies on chromosome 22 [39]. Since the expansion was relatively recent, the duplicated promoter sequences did not have time to diverge, even in neutrally evolving positions. The ability to detect all these architectures in a sample dominated by maternal RNA

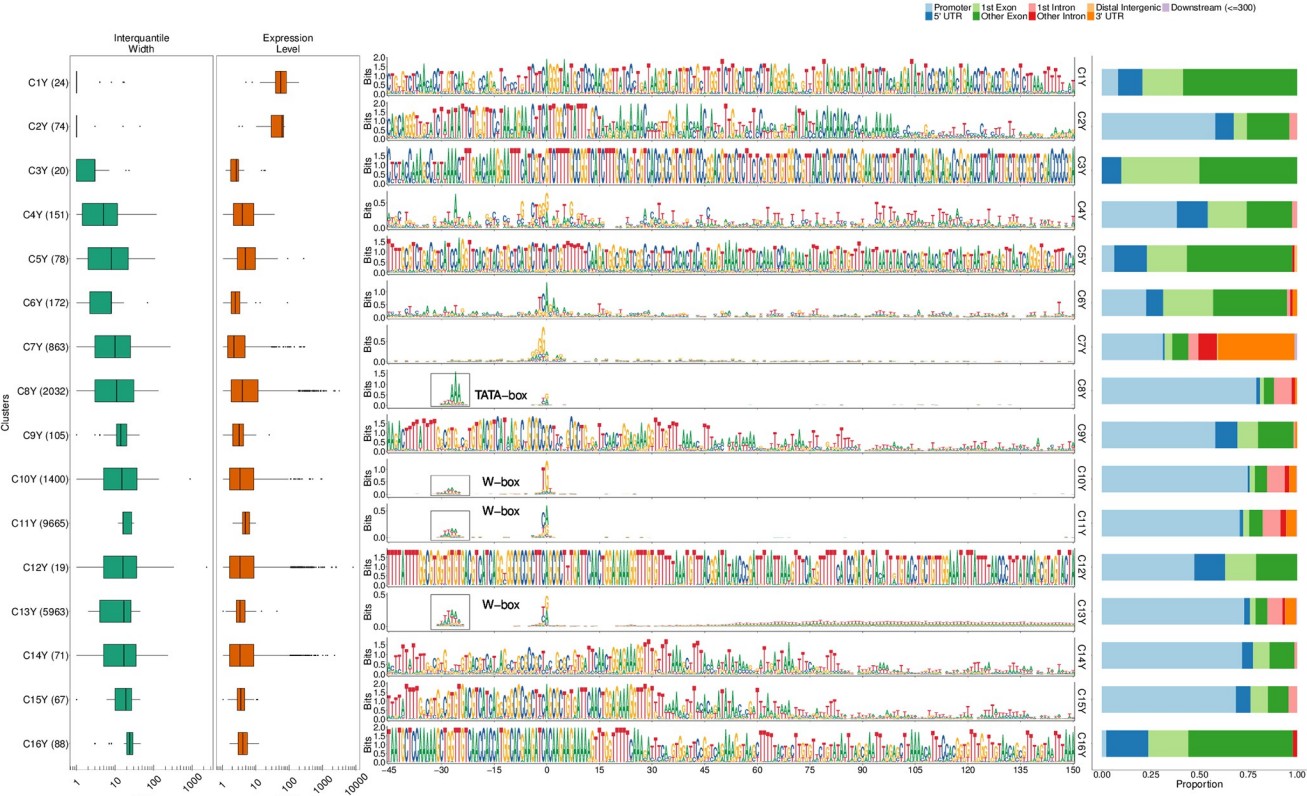

**Fig 9. Clusters and architectures identified by seqArchR for 30% Epiboly/Dome stage.** Sequence clusters arranged by the median IQW of CAGE TCs in seqArchR clusters (shortest on top, broadest at the bottom). Each panel, from left to right: Box and whisker plots of per-cluster IQWs, and TPMs followed by per-cluster sequence logos, and stacked barplots showing proportion of different genomic annotations. TPM, Tags per million.

shows that seqArchR is a powerful tool for characterising regulatory transitions at the promoter level.

Across all clusters at 64 cells stage, the proportion of promoters that are exclusively used at this stage (stage-specific) is very low (Fig AH in S1 Text). This behaviour continues into the 30% Epiboly/Dome stage with exceptions of clusters with promoters from chromosome 4. Contrastingly, at the Prim-6 stage, many new stage-specific promoters get introduced (average 25%).

**Gene ontology of main clusters.** The GO terms enriched for selected clusters are shown in Fig 11B. In the figure, the clusters across different developmental stages are grouped based on the attributes of their architectures. The set of clusters on the left (group with red bar along the horizontal axis) contains clusters with no downstream enrichment of W (A/T) signal *vs* that on the right (blue bar) is those with downstream W signal. The left group has distinctive enrichment for development-related GO terms. These clusters are enriched for different GO terms (Fig 11B). Among the three broad clusters, C12–13Z and C15Z, at Prim-6 stage, C12Z with a well-positioned + 1 nucleosome has different terms enriched in comparison to C13,15Z denoting different utilisation of architectures.

**Temporal successions of motif use in early zebrafish development.** Like in Drosophila, we observe some motif use successions in early zebrafish development as well. Comparing promoter architectures in early to later stages, the shift away from maternal W-box–based architectures is evident in promoter sequences emerging in the 30% Epiboly/Dome stage. By Prim-6

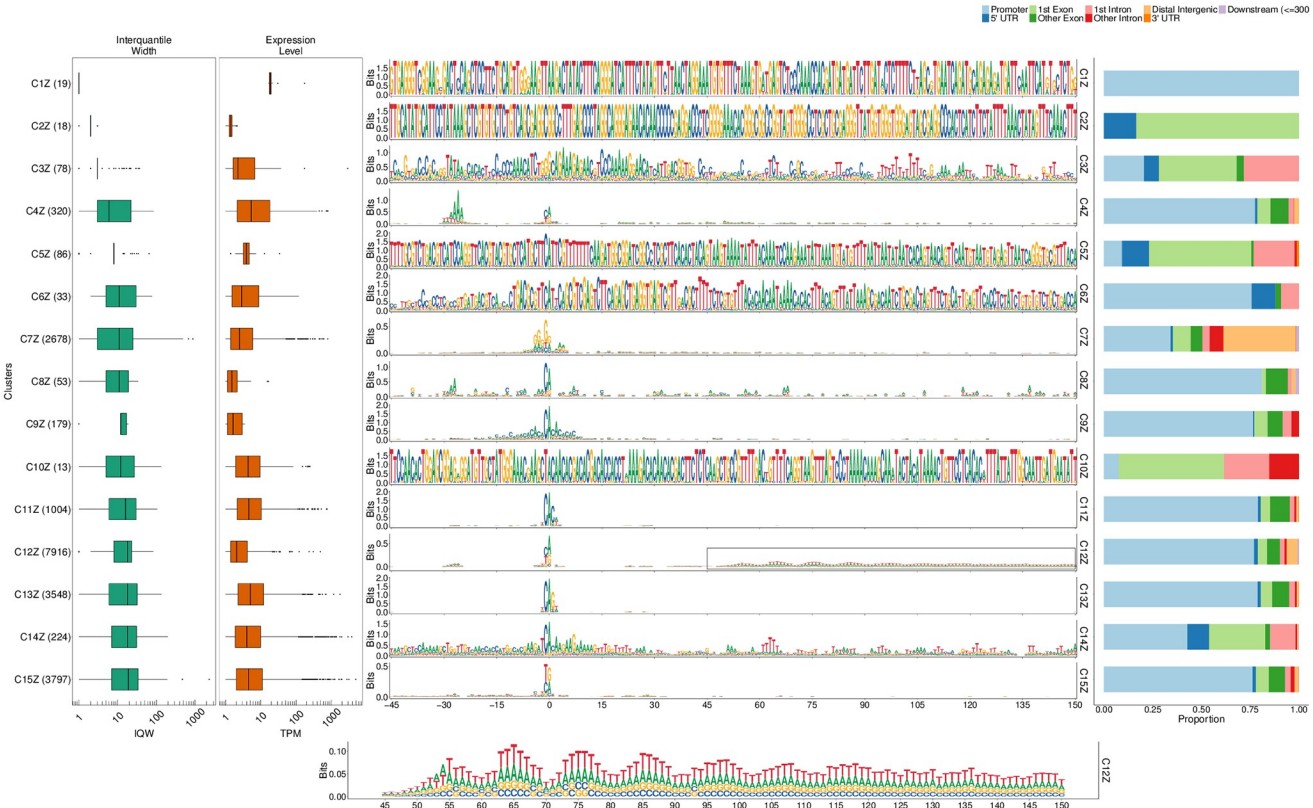

**Fig 10. Clusters and architectures identified by seqArchR for Prim-6 stage of zebrafish development.** Sequence clusters arranged by the median interquantile widths (IQW) of CAGE TCs in seqArchR clusters (shortest on top, broadest at the bottom). From left to right: Box and whisker plots of per-cluster IQWs, and TPMs followed by per-cluster sequence logos, and stacked barplots showing proportion of different genomic annotations. For cluster C12Z, an additional zoomed-in view of the section from 45 bp to 150 bp downstream is shown. TPM, Tags per million.

stage, about half of the promoter sequences have the architecture where dispersed transcription initiation is orchestrated by the first downstream nucleosome (C12Z). Baranasic et al. [35] also reported that the chromatin organisation of promoters between 30% Epiboly/Dome and Prim-6 stages remains stable.

## Human (*Homo sapiens*)

Human promoter sequences are known to be challenging for sequence analysis methods owing to their sequence heterogeneity and, in the case of majority of protein-coding genes, high GC and CpG percentages. To test seqArchR on human core promoter sequences, we gathered CAGE data for different human cell lines and tissues from ENCODE. They were pooled by merging CAGE data for all cell lines (see Fig N in S1 Text). Using a threshold of 1 TPM, about 9500 promoter sequences were analysed using seqArchR.

**Representative promoter windows.** First, we tried the same sequence window as in zebrafish (−45, +150) because we expected from the FANTOM5 analysis (supplementary figure 13 in Andersson et al. [40]) that a majority of CpG-island overlapping and broad promoters will have a stable position of +1 nucleosome and an underlying nucleosome positioning signal, albeit weaker than in zebrafish. Keeping the bound value moderately stringent at $10^{-6}$, the analysis separated architectures that had a TATA-like signal, ca. 30 bp upstream, from the CG-rich architectures (see Fig AC in S1 Text). The inability to identify a stronger, canonical, TATA

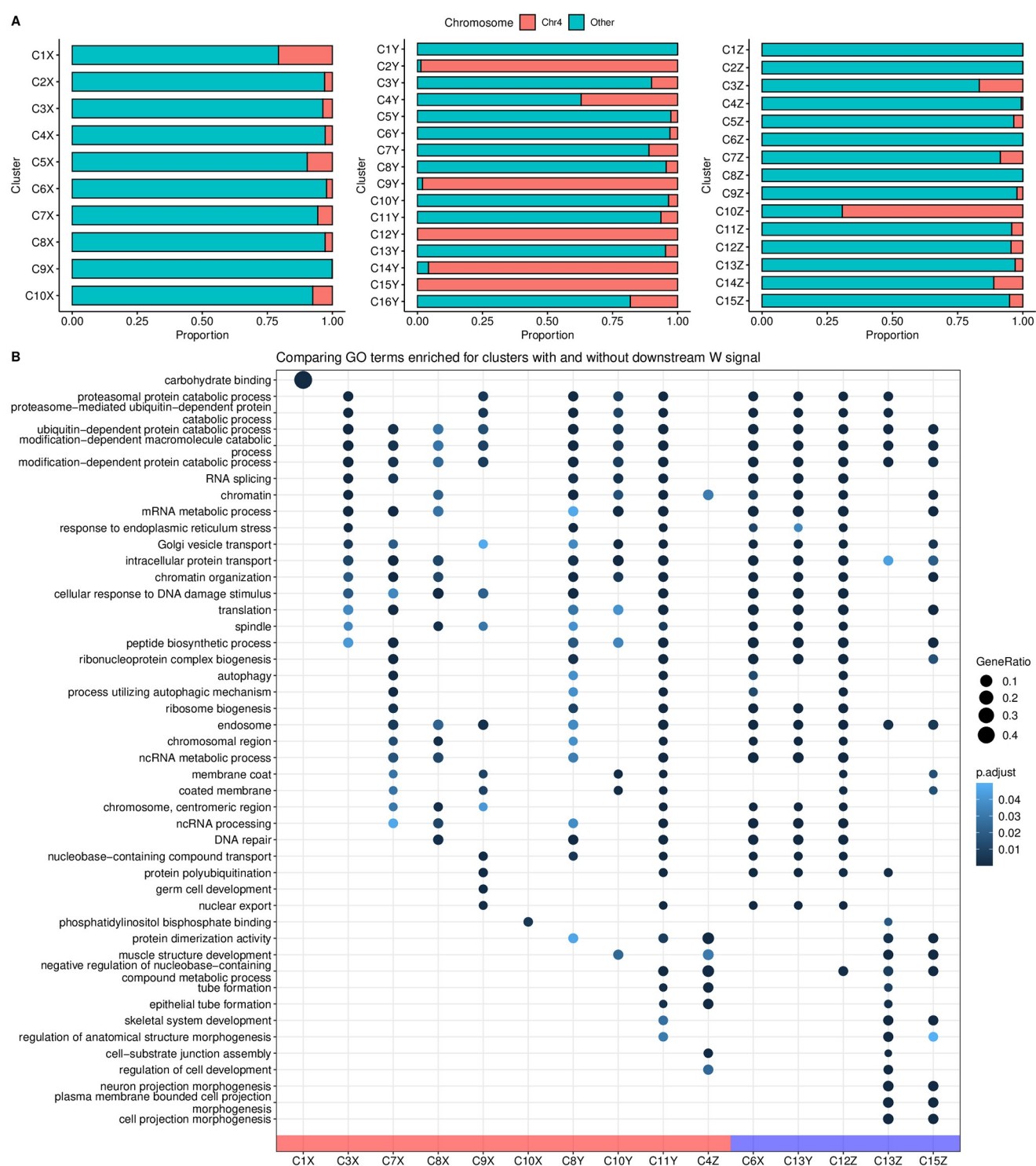

**Fig 11.** (**A**) Genomic locations of per-cluster promoters in Zebrafish developmental stages 64 cells, 30% Epiboly/Dome and Prim-6. (**B**) Comparison of top-5 enriched GO terms for different clusters per stage of zebrafish development. Clusters grouped by architecture attributes: absence *vs* presence of downstream enrichment of W (A/T) signal.

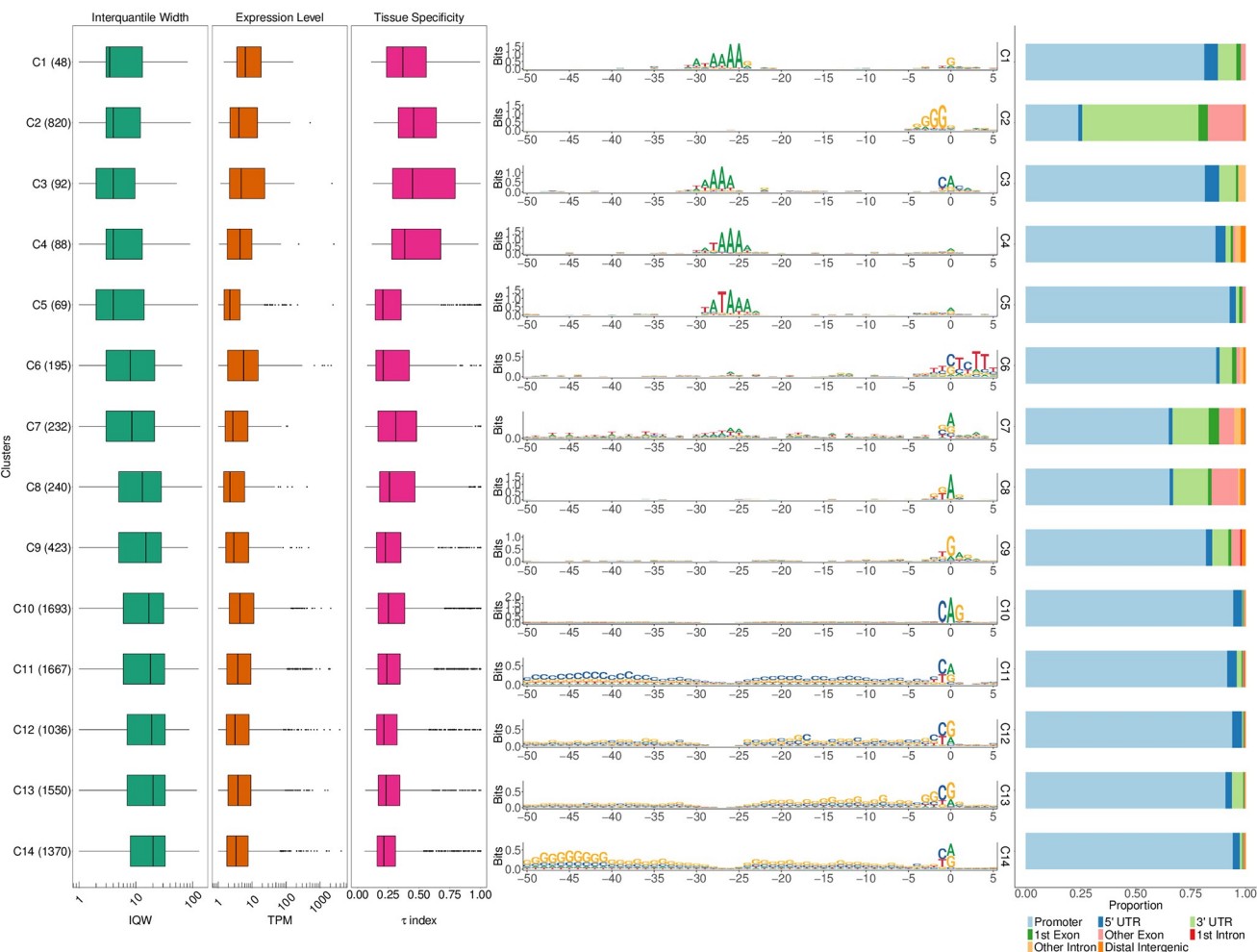

**Fig 12. Clusters and architectures identified by seqArchR for CAGE-derived core promoter sequences from cell lines and tissues of H. sapiens.**
Sequence clusters arranged by the median interquantile widths (IQW) of CAGE TCs in seqArchR clusters (shortest on top, broadest at the bottom). From left to right: Box and whisker plots of per-cluster IQWs, TPMs, and tissue specificity scores ($\tau$) followed by per-cluster sequence logos, and stacked barplots showing proportion of different genomic annotations. TC, tag clusters. TPM, Tags per million.

signal as a separate class is possibly because of the dominant influence of the downstream flanking sequence in the chosen window on identifying clusters with the given bound value. To try to prevent drowning of this TSS-determining motif architecture, we repeated the analysis using flanks of −50 bp and + 5 bp around the dominant TSS. Fig 12 shows the clusters and corresponding architecture sequence logos. Only 297 of these, which is 4.24% of all the core promoter sequences, are observed to have the TATA motifs (Fig 12, clusters C1 and C3-C5). The reason for this low percentage is due to the way the set of promoters was obtained: by pooling CAGE signal from many different tissue and cell types, the signal from tissue-specific genes is diluted, unlike that of broadly expressed genes, bringing a higher proportion of the former under the 1TPM threshold (see S1 Text, section "Diminished tissue-specific signal among the all-pooled CAGE data for H. sapiens.") We used the $\tau$ values as a measure of tissue specificity of promoters [41]). Of these, clusters C1, C3 and C4 have a higher median tissue specificity score than the rest of the clusters. The proportion of housekeeping genes is also extremely low in these clusters (Fig 13B). The top-5 enriched GO terms for each cluster

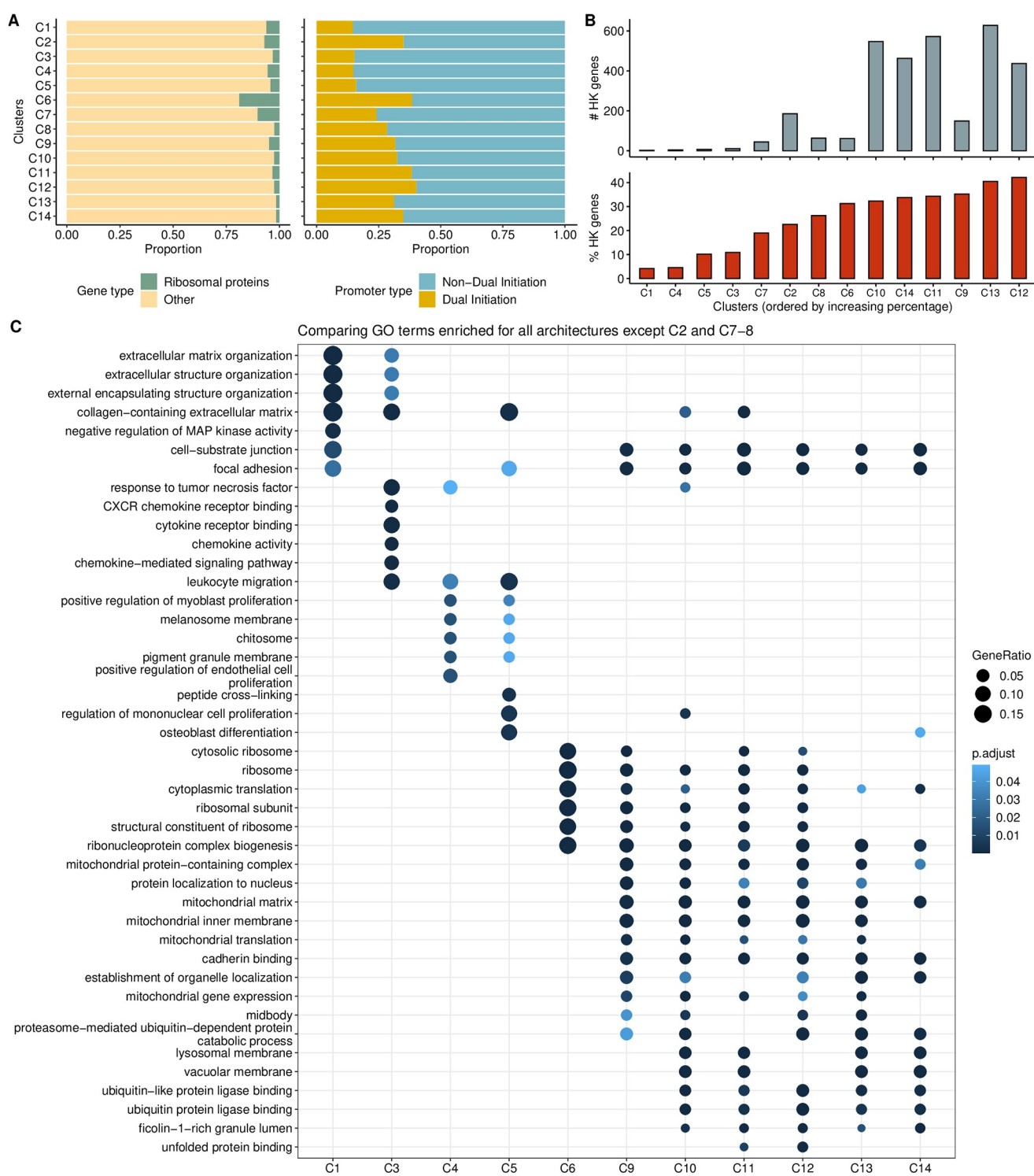

**Fig 13. Additional analyses of seqArchR clusters for H. sapiens. (A)** Per-cluster proportions of ribosomal and other (non-ribosomal) genes (left) and proportions of dual initiation and non-dual initiation promoters (right). **(B)** Proportion of housekeeping (HK) genes in each cluster with clusters arranged in ascending order of % HK genes (bottom) and absolute numbers (top). **(C)** The top-5 enriched GO terms for all clusters except C2 and C7–8 which have more than 25% non-promoter CTSSs. seqArchR cluster names correspond to those from Fig 12.

(excluding C2 and C7–8, which have more than 25% non-promoter CTSSs) are shown in Fig 13C. A clear distinction of enriched terms is observed between tissue-specific/focused architectures and broader architectures.

The `TCT` architecture seen in C6 contains the largest percentage of ribosomal protein genes (Fig 13A). They are also seen in the `TATA` architecture clusters, albeit much fewer in proportion. This is not surprising given that earlier studies have reported that the promoters of these genes in vertebrates are known to harbour `TATA` or `TATA`-like sequences [42–44]. Nepal et al. [25] reported on dual initiation promoters, where `YC` initiation (which includes `TCT`) is intertwined with canonical initiation (`YR`) in metazoans, and also being pervasive in human and Drosophila. It was also shown that dual initiation promoters are often broad in shape owing to many TSSs in vicinity of each other. Our observations here for human promoter architectures are in line with this (see Fig 13B)—proportion of dual initiation promoters is higher in clusters with broad promoters, with cluster C2 (non-promoter CTSSs) and C6 (`TCT`/`YC` architecture) as exceptions.

**Exclusion of initiator sequence and subsequent analysis.** We envisaged that, at least in vertebrates, because the initiator is degenerate but still adhering to the `YR` consensus, it may lead to splitting clusters by the four possible `YR` dinucleotides, a split uninformative of the functional promoter architecture. Thus, to test this impact of the initiator on the clustering, we compared the clusters obtained with and without the initiator sequence. Refer to Fig AM in S1 Text for the following. The results show some interesting observations. In most of the clusters, a large proportion of promoters stay together when with or without the initiators. Examples of such behaviour are clusters C2–5, C8–9, C11–13, C17,19–20 (left panel; with initiator), and approximately retain their shape-based rankings between the two. In some clusters, such as C14–16 (with initiator) where the flanks show slight `GC` enrichment, a good chunk of these promoters get distributed across some narrow and broad architecture clusters, implying that they clustered together due to the initiator sequence `TG`.

## Discussion

Promoter sequence architectures are much more than just individual sequence motifs lying anywhere within the promoter region. In some instances, these sequence motifs must occur in tandem, at fixed (relative) distances from one another. In addition to motifs, the sequence composition also plays an important role. This gives rise to a diverse set of sequence architectures even within a single species. Organisms are known to use different core promoter grammars in different regulatory contexts, such as different stages of embryogenesis, e.g., early *vs* late, or for different functions, e.g., tissue specific *vs* housekeeping functions. Moreover, there is diversity even across organisms. Therefore, there is a need for *de novo* approaches to explore and study the diverse promoter sequence architectures across species and organisms.

We present seqArchR, an approach for *de novo* identification of sequence elements based on non-negative matrix factorisation. seqArchR can be used for studying promoter sequences aligned by the position of their TSSs. seqArchR's chunking-based iterative algorithm can handle scenarios of complex interactions between sequence (motif) elements. Experiments on simulated DNA sequences show its ability to identify strong and/or weak artificially planted sequence elements. seqArchR is recognised as much faster (ca. 20–30x) than No Promoter Left Behind, the current state-of-the-art approach for *de novo* identification of promoter sequence architectures.

Different organisms are known to pose different challenges for sequence analysis. For example, Drosophila core promoter sequences harbour multiple motifs with high information content, e.g., `TATA`-box, DPE, Ohler motif 5, embedded in otherwise low information regions.

In comparison, most promoters in some other organisms lack such motifs. For example, most zebrafish zygotic promoters have only a YR initiator and WW periodicity downstream demarcating nucleosome position [5]. Only a small percentage of core promoters in human have a TATA-box, while the majority of their promoter sequences have a higher GC content throughout. seqArchR can detect all types of motifs and non-motifs–based sequence elements. We managed to retrieve this information by processing promoter sequences across the three aforementioned organisms. Depending on the nature of sequence elements, some parameters for seqArchR, namely the stability bound, number of iterations, and per-iteration collation decision, need to be set appropriately. Adjusting the *stability bound* value enables adjusting the sensitivity of seqArchR to identify even weak sequence features that may be enriched in very few sequences. As discussed in the Methods section, a good choice of number of iterations and the per-iteration collations can help identify overall defragmented clusters while still keeping clusters with minor positional variations separate. Indeed, the ability to tune seqArchR in this way makes it work seamlessly across organisms.

seqArchR is developed to discover the promoter sequence components tied to the precise transcription initiation sites identified using CAGE experiments. Therefore, it treats absolute position information of sequence features as an important characteristic of any architecture. We note that, for this approach to work, nucleotide-resolution data such as that from CAGE is required. For instance, ChIP-seq signal is not sufficiently high-resolution to be useful for aligning the sequences for NMF analysis. We envisage that ChIP-exo or ChIP-nexus data, which yield high-resolution information of transcription factor binding sites, can be analysed informatively with seqArchR.

## Is seqArchR another motif finder?

seqArchR has some similarities with general motif finding method, and important differences. Because seqArchR analyses sequences that can contain enriched motifs, it is able to identify motifs, albeit only positioned, i.e., aligned ones. However, seqArchR uses the complete sequence to directly infer clusters of sequences with common features. This is affected by the composition of the whole sequence, including the flanks of the motif, if present. This is unlike any motif finding approach, which looks for statistically enriched words of pre-determined size (motif length) (usually 6–15 bp and only seldom larger). Another important distinction is that seqArchR does not require/use any background (sequence) information as motif finders do. This limits seqArchR's ability to find motifs enriched in set A *vs* set B which is a common usecase scenario for motif finders (i.e., differential motif enrichment). In this case, seqArchR can still be used to independently analyse set A and set B and then compare the identified features. In a nutshell, seqArchR is a clustering approach, where as typical motif finders can be considered classification-based.

seqArchR could be further improved in the following ways. It currently uses the same set bound value for every chunk/cluster in every iteration. It may be beneficial to extend this to adaptively change the bound value per chunk/cluster in any iteration. In terms of its algorithm, the approach for identifying overfit clusters can be improved. Furthermore, candidate approaches to automate the decision to collate after every iteration can be explored. Also, the current approach for collation can be improved to reduce manual intervention. In its current form, seqArchR only reports complete profiles of the same length as the input sequences. We want to extend it to be able to extract motifs from complete profiles like TF-MoDISco [45].

## Conclusions

seqArchR is a *de novo* approach for discovering and clustering promoter sequence architectures. With seqArchR, we demonstrated the ability of NMF to simultaneously discover clusters of sequences with diverse positioned sequence motifs. seqArchR seamlessly identifies promoter architectures across organisms which are characterised by sequence motifs or the underlying sequence compositions. It is much faster than the current state-of-the-art, NPLB. seqArchR is made available as an R/Bioconductor package http://www.bioconductor.org/packages/seqArchR and a website accessible at https://snikumbh.github.io/seqArchR.

## Materials

All experimental CAGE datasets analysed in this study were obtained and processed as follows. CAGE data for embryonic developmental stages in Drosophila melanogaster were obtained from Schor et al. [17]. Out of different strains/samples available, sample 'RAL_28' was randomly chosen. CAGE data for three transitions in embryonic development, namely 2–4h, 6–8h and 10–12h after egg laying, was processed using CAGEr v2.0.2. See this Rmarkdown document for more details.

CAGE data for the embryonic developmental stages in Zebrafish were obtained from Nepal et al. [18]. Out of all stages ranging over maternal to zygotic transition, we selected three, namely 64 cells, 30% Epiboly or Dome, and Prim-6 for this study. These were processed using CAGEr v1.20. See this Rmarkdown document for more details.

For Homo sapiens, all cell lines-specific CAGE data was obtained from ENCODE consortium which was made available previously as an R package `ENCODEprojectCAGE_1.0.1.tar.gz` available at http://promshift.genereg.net/CAGEr/PackageSource/. For this study, data from all cell lines was merged and processed using CAGEr v1.20. Library sizes of all cell lines that are included are shown in Fig AJ in S1 Text.

The list of housekeeping genes in humans was obtained from Eisenberg and Levanon [46]. The $\tau$ index, first introduced by Yanai et al. [41]), for all human genes were obtained from Palmer et al. [47]. Palmer et al. used the GTEx data to compute the $\tau$ index of tissue-specificity for all human genes [47]. A value closer to zero indicates equally expressed across included tissues in the GTEx data and a value closer to 1 indicates expressed in one tissue.

The list of dual initiation promoters in humans was obtained from Nepal et al. [25].

## Methods

NMF seeks a low-rank decomposition of any given matrix $V_{p \times n}$ which only has non-negative entries. Here, $n$ is the number of samples and $p$ the number of attributes. We note that attributes can also be referred to as features and $V$ as data matrix.

$$V_{p \times n} \approx W_{p \times k} H_{k \times n} \qquad k \ll p; \quad V, W, H >= 0 \tag{1}$$

The matrix $H_{k \times n}$ records the coefficients of the $n$ samples in the $k$-dimensional representation, $k \ll p$. The matrix $W_{p \times k}$ provides the loadings of the original attributes, $p$, in the $k$-dimensional space. $W_{p \times k}$ is called the basis matrix (and we refer to each column of $W$ as a basis vector), and $H$ the coefficients matrix. All entries in $W$ and $H$ are non-negative. Thus, NMF achieves dimensionality reduction, and can be further used for clustering using information in the coefficients matrix. Here, it is assumed that the lower-dimensional representation obtained using NMF corresponds to different clusters present in the data, and the clusters are characterised by different groups of attributes. Due to non-negativity, NMF lends an intuitive parts-

based representation in many applications where negative values do not have an intuitive meaning.

The basic idea of seqArchR is to process a given set of sequences with NMF. These sequences are represented in the form of a one-hot encoded matrix, whose each column corresponds to a sequence and the sequence features are along its rows. In the following, we describe in detail the input to seqArchR and its chunking-based iterative algorithm. Together with it, information on the relavent parameters from the corresponding R package seqArchR is also given.

## Input

The input to seqArchR is a matrix of one-hot encoded representation of DNA sequences of same length, *L*. For each DNA sequence, its mononucleotide profile is one-hot encoded as follows. The four channels in its one-hot encoding—due to DNA alphabet, $\Sigma$ = {A, C, G, T} —are concatenated to produce a single column vector representation per sequence. The size of this vector is four times the length of the sequence itself. Thus, for an input matrix representing mononucleotide profiles of *n* sequences, its dimensions are $(4L \times n)$. All entries in this matrix are non-negative, or more specifically, 0 or 1.

Similarly, the dinucleotide profile of a sequence can be one-hot encoded. The input matrix dimensions in this case are $(4^2L \times n)$. It is known that dinucleotide profiles of DNA sequences hold more information than mononucleotide profiles [48, 49]. All results reported in this article are obtained using dinucleotide profiles.

As shown in Fig 2B, in the decomposition of the one-hot encoded input matrix $V_{p \times n}$ to $W_{p \times k}$ and $H_{k \times n}$, $W_{p \times k}$ assigns scores to each (di)nucleotide at a given position. Each column of $W_{p \times k}$, a single basis vector, captures combinations of various nucleotides at characteristic positions in sequences and can be said to represent an architecture. As an example, consider architectures I-IV in Fig 2B. All important features ((di)nucleotide-position pairs) are captured at once in a single basis vector of *W*. $H_{k \times n}$ gives the coefficients for each sequence on all *k* basis vectors. These coefficients are relative in nature. When a particular sequence has features characterised by basis vector A but not those by basis vector B, it gets assigned a higher coefficient for basis vector A than for B. This information can be used for soft clustering of sequences. In this fashion, coefficients for all *n* sequences can be interpreted. Thus, NMF can simultaneously learn the complex interdependencies and interactions between sequence features *de novo*, and also provide soft clustering of input sequences.

Given a set of DNA sequences as a FASTA file or a `Biostrings::DNAStringSet` object [50], the R package seqArchR has provision to generate one-hot encoding of mono- as well as dinucleotide profiles of sequences. This can then be used as an input.

## Algorithm

seqArchR's algorithm for *de novo* identification of clusters and architectures among the given sequences is depicted pictorially in Fig 2A. Here, we describe the algorithm in full.

1. **Chunk the given sequences**. Given the input matrix, the total collection of sequences is divided into smaller chunks (subsets). These *chunks* are of a pre-defined size that can be set by the user (parameter: `chunk_size`).
   Note on terminology: With every chunking operation on a set of sequences, the resulting chunks of sequences are referred to as *inner* chunks, and the parent set of sequences are referred to as the *outer* chunk. At the first iteration, we have one outer chunk (the complete

set of input sequences), and many inner chunks. At subsequent iterations, we expect more than one outer chunks, and one or more inner chunks per outer chunk.

2. **Independently process each inner chunk with NMF as follows**.

(a) **Finding the appropriate number of clusters/basis vectors ( *model selection*)**: The number of basis vectors is given by $k$ in Eq (1). We test a range of values for $k$ (parameters: `k_min` and `k_max`) from which the optimum value is selected based on either of the two quantitative criteria mentioned below (parameter: `mod_sel_type`).

- *Model selection using stability of obtained basis vectors*. Wu et al. [51] proposed and used the Amari-type distance that for a given value of $k$, measures the instability of the identified basis vectors over multiple runs. They termed the set of basis vectors $R$ learned in a single run as a dictionary. With $R_m$ dictionaries from $m$ runs, the dissimilarity between any two dictionary pairs $R$, $R'$ is given as

$$diss(R, R') = \frac{1}{2K} \left( 2K - \sum_{j=1}^{K} \max_{1 \le k \le K} C_{kj} - \sum_{k=1}^{K} \max_{1 \le j \le K} C_{kj} \right). \tag{2}$$

The instability of the set of identified $k$ basis vectors is the average dissimilarity of all dictionary pairs over $m$ multiple runs as computed by Eq (2). Here, $C_{K \times K}$ is the cross-correlation matrix of basis vectors (columns of $W$) from $R$ and $R'$ while $j$ and $k$ are column and row iterators respectively. Wu et al. [51] used this for studying spatial gene expression patterns.

With increasing values of $k$, the instability increases. seqArchR sets a bound on the instability. The maximal value of $k$ whose instability is lower than the bound is selected. In the R package, the `bound` parameter can be set by the user. Based on experiments on simulated data and various biological datasets, the recommended value of `bound` is one of $10^{\{-6,-7,-8\}}$. Higher powers (e.g., $10^{-8}$) are more stringent and lower ones (e.g., $10^{-6}$) more lenient. The user can choose values beyond this range as may be suitable for a dataset. Note that some weaker clusters may not be identified with stricter bound values, in which case lenient bound value is recommended. The default value of *bound* is set to $10^{-6}$ (lenient). The default/recommended number of bootstrapped iterations when using this approach for model selection is 100 (parameter: `n_runs`).

- *Model selection using bi-cross-validation*. $r$-fold bi-cross-validation (bi-CV) approach was proposed by Owen and Perry [52] and recently used by Eng et al. [53]. This approach tests the re-construction accuracy of the NMF solution for each value of $k$ and chooses the one that maximises it. Multiple bootstrapped iterations of NMF are performed for each value of $k$. The one with the maximum average re-construction accuracy over all bootstrapped iterations is the best one. We further apply the one-standard error rule to choose the most parsimonious model: the smallest value of $k$ for which the reconstruction accuracy is no lower than the one-standard error of the best performing $k$ [54].
The default/recommended number of folds and bootstrapped iterations when using this approach is 5 and 500 respectively (parameter: `cv_folds` and `n_runs`). Note that the multiple runs are performed per-fold. The effective number of iterations thus performed is 2500 per value of $k$.

The stability-based model selection procedure is the recommended approach since it is faster owing to the smaller number of bootstrapped runs required per value of $k$.

(b) **Obtaining clusters**: We then obtain the final NMF solution from multiple runs using the chosen number of basis vectors. We use the coordinate descent solver [55] with Nonnegative Double Singular Value Decomposition (NNDSVD) initialisation [56] which encourages sparsity. Matrices $W$ and $H$ from the run achieving the best re-construction accuracy are collected. The set of sequences in the *inner chunk* is then partitioned into as many clusters as the number of basis vectors using the coefficients in matrix $H$ (Eq (1)). Each sequence is added to the cluster corresponding to the basis vector for which it has the highest coefficient.

(c) **Accounting for overfitting**. Either of the model selection approaches can suffer from overfitting. This leads to identification of clusters where too many features of only few sequences from the cluster are deemed as important. The coefficients assigned to these few sequences for this architecture are very high in comparison to other sequences in the cluster. Thus, the distribution of the coefficients for all sequences in such a cluster shows a large variation. seqArchR detects such cases by default and controls for them by re-merging them with the parent cluster. For example, if a (parent) cluster is divided into three clusters of which the second is adjudged as an overfit cluster, it is merged back into the first (considered as the parent) of the three clusters.

3. **Collate similar clusters (if any) from different inner chunks**. Among the clusters obtained from the various *inner* chunks, we collate those that are similar into one. This is done by clustering similar basis vectors using hierarchical clustering. The distance and agglomeration method used for hierarchical clustering can be set by the user (parameters: `result_dist` and `result_aggl`). One set of choices here are applicable for all iterations.
   Based on various computational experiments, we recommend using ward.D linkage with Euclidean or correlation distance. Note that using any combination of distance and agglomeration method only affects collation of clusters and does not affect per-chunk cluster/architecture identification by NMF.

4. **Re-iterations**: Designate all obtained clusters as separate *outer chunks*. Each *outer chunk* is treated as an independent collection of sequences. Steps 1–3 are repeated per outer chunk separately with the following caveat. As a result of the collation, if any new outer chunk is much larger than the user-specified `chunk_size`, it is further divided into 'inner' chunks of user-specified `chunk_size` as per step 1 and processed with steps 2 and 3.
   The designation of obtained clusters as outer chunks and processing them over the next iteration is handled automatically. seqArchR performs a user-specified number of iterations in this fashion (parameter: `total_itr`).

The clusters identified in the last iteration of seqArchR are collated and reported as the final result. This collation of clusters for final reporting is treated as separate from the per-iteration collation.

## Important note on collation of clusters and number of re-iterations

Keeping in mind the exploratory nature of any clustering analysis, the decision to collate similar clusters from different chunks (step 3) and number of iterations (step 4) have no defaults because they are dependent on the nature of the data.

**Suppressing or deferring collation.** In addition to the choice of the distance measure and the agglomeration method to be used for collating clusters using hierarchical clustering, the per-iteration decision to collate clusters is also left to the user (parameter:

set_ocollation). One can suppress collation for a particular iteration or all of the specified number of iterations. For instance, if three iterations are to be performed, the user can choose to collate clusters only for the second iteration. Deferring collation is a useful strategy, especially in scenarios where the sequence features/motifs have minor positional variations. For example, in Drosophila promoter sequences, where TATA-box is observed anywhere between 32 to 28 bp upstream of the TSS, deferring collation to the second iteration is helpful to detect these minor positional differences right away. In fact, in all experiments reported here, except those with simulated data, we defer collation to the second iteration.

**Number of iterations.** The appropriate number of iterations depends on the nature and number of sequence architectures. It is almost always a good idea to perform at least three iterations or at least one iteration after a deferred collation exercise when analysing real promoter sequences.

## Curation of collated clusters

Another point to note is that depending on the choices for distance measure and agglomeration method, programmatic collation of basis vectors with hierarchical clustering can sometimes combine clusters which are perhaps better off separate. Indeed many studies require some manual curation when combining position weight matrices (PWMs) with hierarchical clustering [57] Therefore, we recommend that the clusters from the last iteration be left uncollated (by appropriately setting the parameter set_ocollation to FALSE for the last iteration). The final output from seqArchR will still be a collation of clusters from the last iteration which, as noted above, is treated as independent of per-iteration collation. Keeping clusters from the last iteration uncollated provides the user with a choice to collate the clusters in a custom manner and curate them when necessary. The users can either use built-in functions from seqArchR package or any other custom script. All results reported in the manuscript follow this strategy (except simulated data results). Figs F, G, and H in S1 Text show how this process was carried out for D.melanogaster; Figs L, M, and N in S1 Text show this for D.rerio; and Fig Q in S1 Text shows this for H.sapiens.

Thus, seqArchR requires no prior information or specification about the nature of architectures. It can distinguish sequences with motif-based architectures from non-motif-based architectures. If there are motifs, it does not require any information on the expected number of motifs or motif lengths or if any of the motifs have gaps (and the size of the gaps), or if some of them work cooperatively resulting in a combinatorial interplay.

## Rationale for the chunking-based algorithm

As discussed in the Results section, promoter sequence architectures often harbour only slightly varying sequence elements at the same position in all sequences. For example, the canonical YR initiator—case of same length motifs overlapping in occurrence positions, and Inr CA overlapping with the first CA of Ohler1 motif in Drosophila. A chunking-based approach makes it relatively easier to select the best model by alleviating the above mentioned co-occurrence cases and handling fewer sequences at once. Fig D in S1 Text shows the architectures of clusters, identified among Drosophila promoter sequences from modENCODE, using NMF with stability-based bound criterion for model selection in the first iteration. It can be seen that, the first set of identified clusters (across chunks) are majorly governed by initiator elements.

Additionally, for larger chunk-sizes, the memory usage increases linearly (see Fig B in S1 Text). Therefore, having the ability to process very large datasets one chunk at a time enables

processing them with a reasonably smaller memory footprint. Qualitatively, a very small chunk size can lead to a fragmented set of final clusters.

## Visualising promoter architectures as sequence logos from NMF results

**Sequence logos from candidate sequences.** All sequence logos presented in this paper are obtained as follows. For each cluster identified by seqArchR, the sequences assigned to that cluster are used to obtain a position frequency matrix (PFM), which is then visualised as a sequence logo showing information content (bits) at each position in the consensus sequence [58].

**Sequence logos from NMF basis vector (weights).** Alternatively, one could visualise the NMF basis vector for each identified cluster also as a sequence logo. Recollect that the NMF basis vector is nothing but the weights assigned to each feature in the $W_{p \times k}$ matrix in Eq (1). Each basis vector is first transformed into a position weight matrix (PWM)-like two-dimensional matrix with dimensions $4 \times L$ if mono-nucleotide features are used, or $16 \times L$ when dinucleotide features are used. For dinucleotide features, the $16 \times L$ matrix is collapsed into a $4 \times L$ matrix. Each column pertaining to the sequence positions is normalised to 1. This PWM-like matrix is then visualised as a sequence logo or a heatmap. See examples in the seqArchR vignette on Bioconductor.

We note that these PWM-like matrices are analogous to the contribution weight matrices identified by TF-MoDISco [45] for deep learning models, albeit for complete sequences instead of just motifs or motif positions in the sequences.

## Method availability

seqArchR is made available as an R/Bioconductor package with a GNU General Public License version 3 (GPL-3). It can be accessed at: http://www.bioconductor.org/packages/seqArchR The website documenting the package is also accessible at https://snikumbh.github.io/seqArchR and the source code at https://github.com/snikumbh/seqArchR. seqArchR was previously named archR (version 0.1.8). This is also the version used for experiments and results reported in this manuscript. It is deposited at Zenodo with DOI 10.5281/zenodo.5055408 [59]. Following version 0.1.8, there have been no changes to the algorithm.

## Reproducibility of results presented in this manuscript

All results presented in this manuscript can be reproduced. Source code for all analyses and figures is available on GitHub at https://github.com/snikumbh/reproducible-seqArchR-manuscript, as a `remake`-based reproducible pipeline [60]. `remake` is a Make-like build management system for R. When the R package `remake` is installed, running `remake::make()` at the R prompt inside the project folder will install the required R libraries/packages, download the data folder from Zenodo (caution: large size), and knit the organism-specific Rmarkdown files to produce the analyses figures. For more detailed help/instructions, please see the README file in the repository.

Specifically, seqArchR clustering results for Drosophila CAGE data from Schor et al. [17] can be reproduced using the R script archR-on-drosophila-schor2017.R and downstream analyses using archR_dm_schor_figures.Rmd. Similarly, R scripts and Rmarkdown files are available for reproducing Zebrafish and Human results.

The results from NPLB on Chen et al. [16] CAGE data (S1 Text, section 2) can be reproduced by running `promoterLearn` on the input FASTA files like `promoterLearn -f <input_file> -o <output_directory>`.

## Supporting information

**S1 Text. Supporting information document.**
(PDF)

## Acknowledgments

We thank Leonie Roos for processing earlier versions of Drosophila and Zebrafish CAGE datasets, and members of the Computational Regulatory Genomics group for discussions through out the project and feedback on the manuscript.

## Author Contributions

**Conceptualization:** Sarvesh Nikumbh, Boris Lenhard.

**Formal analysis:** Sarvesh Nikumbh.

**Investigation:** Sarvesh Nikumbh, Boris Lenhard.

**Methodology:** Sarvesh Nikumbh.

**Software:** Sarvesh Nikumbh.

**Supervision:** Boris Lenhard.

**Validation:** Boris Lenhard.

**Visualization:** Sarvesh Nikumbh.

**Writing – original draft:** Sarvesh Nikumbh, Boris Lenhard.

**Writing – review & editing:** Sarvesh Nikumbh, Boris Lenhard.

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
