## [Decision Letter · Decision Letter 0]

31 May 2023

Dear Dr Nikumbh,

Thank you very much for submitting your manuscript "Identifying promoter sequence architectures via a chunking-based algorithm using non-negative matrix factorisation" for consideration at PLOS Computational Biology.

As with all papers reviewed by the journal, your manuscript was reviewed by members of the editorial board and by several independent reviewers. In light of the reviews (below this email), we would like to invite the resubmission of a significantly-revised version that takes into account the reviewers' comments.

We cannot make any decision about publication until we have seen the revised manuscript and your response to the reviewers' comments. Your revised manuscript is also likely to be sent to reviewers for further evaluation.

Sincerely,

Denis Thieffry, PhD

Academic Editor

PLOS Computational Biology

William Noble

Section Editor

PLOS Computational Biology

Reviewer's Responses to Questions

**Comments to the Authors:**

Reviewer #1: Review on the article “Identifying promoter sequence architectures via a chunking-based algorithm using non- negative matrix factorisation” by Nikumbh and Lenhard

In this article, the authors present a method to detect promoter motifs using an approach based on non-negative matrix factorization. Promoter sequences are summarized using a one-hot encoding in which all dinucleotides are considered and grouped into several clusters using a multi-step clustering procedure. The authors validate the algorithm using simulated data and show its superior performance with respect to state-of-the-art methods. One advantage of the presented method, seqArchR is the speed of execution. They then apply the method to CAGE data from different organisms and show that they can recover known promoter motifs and related them to specific functional groups. seqArchR can also detect novel motifs, such as in Drosophila, indicating its high sensitivity.

The article is well written, and the results are convincing. The complex algorithm is well described in the methods. I have a few questions on some of the results:

1. It is not clear how the sequence logos are built from the results of the NMF. This should be described more in detail.

2. Some logos seem to represent fixed motifs (for example C1X, C2X in figure 4), as compared with most other logos which only highlight specific shorter motifs within the promoter sequence. I was wondering why some of the logos have this property. It is probably related to the small number of sequences in the cluster, however Fig S8 seems to show example of small clusters which do not show this fixed motif logo. Could the authors comment on this, and whether these “rigid” motifs should be removed from the output?

3. Selection of the factorization rank is a critical aspect in NMF decomposition. The authors present two heuristic methods in the Methods section describing the algorithm. The model selection using bi-cross-validation considers the reconstruction accuracy of the NMF. This sounds to me like the criteria of the Froebenius error measuring the reconstruction error, which generally favors larger values of k. Could the authors comment on possible biases towards higher factorization ranks?

4. Related to the previous comment, how sensitive is the choice of k in the seqArchR method? How much do the results vary with different values of k?

Reviewer #2: This paper describes a (i) new method to classify DNA sequences of fixed length aligned on an experimentally defined position, and (ii) an application of this new method to promoter sequences of three animal species (aligned on TSS). Overall, this is a paper of high scientific quality, clearly written, and presenting interesting biological results. To me, the usage of non-negative matrix factorization to find subclasses in DNA sequences is novel, and I consider it a pretty smart choice.

The methods are well described, software is available from git hub, and the input data to the computational workflows have been deposited in Zenodo. The study thus meets high standards of reproducibility.

I have a few requests for minor modifications.

Results Section:

My understanding (based on Methods and Supplementary material) is that the architectures shown by sequence logos in the main manuscript were obtained by manual curation and post-processing of automatically generated architectures returned by SeqArchR (shown in supplementary Figures). This should be explicitly and clearly stated in the Results section.

The DPE element is barely visible in the sequence Logos of Drosophila architectures from 2-4h AEL (Fig. 4) and 10-12 AEL transitions. The only architecture with a clearly visible DPE element (C1Y) comes from transition 6-8h AEL. This is surprising and inconsistent with the notion that DPE promoters are characteristic of developmental genes. Please comment.

Page 11: "In C8Z, there are two initiators, the canonical CA and full Inr, their A+1 3nt apart". The concept of a "full Inr" should be introduced before in the text, ideally by sequence motif and literature reference.

Page 11: "TATA and DPE sequence elements are known to mutually exclusively pair with the Inr for architectures characteristic of tissue-specific and developmental genes respectively" If possible, justify this statement by a reference.

Methods:

Page 22: "Library sizes of all cell lines that are included are shown in Supplementary Figure 4." Should probably be Supplementary Figure 16.

Page 23: Textually define the variables K, C_jk and C_kj occurring in Equation 2.

Below are some comments and suggestions, which the authors may or may not consider at revision stage. I emphasize that they should feel completely free in their choice.

The high information content Logos are intriguing. I find it interesting that some of these logos correspond to histone promoters and one to genes encoding mir-430. I would be curious to know more about other information rich logos, for instance C4Z from zebrafish.

I have some doubts about the genuine difference between a TATA-box and W-box motif. Since this analysis requires exact superposition for sequence logo generation, W-box-like motifs obtained in this study may simply represent canonical TATA-boxes shifted by +/-2 bp.

A different method based on probabilistic mixture modelling was used in a recent study with partially overlapping objectives (Dreos et al. PLoS Comput Biol. 2021). The authors may comment on the differences in the methodological approaches and/or results obtained for Drosophila and human promoters. One striking difference concerns the Drosophila DPE promoters. According to this other study, more than one third of Drosophila gene promoters conform to an architecture featuring a full Inr and a clear-cut DPE motif (resembling Drosophila C1Y in this paper). Reasons for this discrepancy could be discussed (different gene subsets, upstream Cage data processing method, architecture discovery algorithm, etc).

Reviewer #3: Nikumbh and Lenhard present a new method for clustering promoter sequences arising from high-resolution experimental data such as CAGE. They use non-negative matrix factorisation for this purpose. The method is timely, given new technologies emerging that give such high-resolution data.

A few issues, which should be clarified:

* Citation needed on page 3 for "Sharp promoters are associated with tissue-specific genes, while broad promoters are associated with constitutively expressed genes and a subset of tissue restricted ones."

* Figure 1: Caption mentions "across organisms", but that is not indicated in the figure.

* Figure 2: If one is using one hot encoding for dinucleotides should the length of the vector not be 4x4x(L-1)? The figure, as I understand it, shows an AA at position 2 and at position 4 in the first sequence. Does that not mean the third position should also have an AA? Is the Vpxn matrix a real representation? It should be, ideally.

* Table 1: Authors should mention the position where the AT period starts in cluster A. The reader can get the wrong impression that the ATs start at arbitrary positions, but occur ever 10 bases.

* What commands should be run to reproduce the results, in case of both seqArchR and NPLB? These should be stated in the Methods.

* In Fruit fly, authors do not identify the TATA motif in two of the time points. The explanation given may be correct, but one needs to ensure that there really is no TATA motif in those promoters. If you do a scan for the TATA motif is it really not present? Do none of the iterations find it? Would increasing the number of clusters find the TATA motif?

* Not sure if "successions of motifs" is common terminology. It appears twice and only in the subheadings. It should be described somewhere, or change it.

* On page 19, the authors state "To try to prevent drowning of this TSS-determining motif architecture..." This needs to be qualified by a figure with the original results of using the full sequence length. Furthermore, instead of going only after the TATA motif, why not also look at the downstream regions where the claim is that there exists "dominant influence of the downstream flanking sequence"? More explanation/analysis is necessary here.

* "We speculate that the reason for this low percentage could be due to the way the set of promoters was obtained: by pooling CAGE signal from many different tissue and cell types where signal from tissue-specific genes is diluted, unlike that of broadly expressed genes, bringing a higher proportion of the later under the 1TPM threshold. " This can be verified. If you take a single tissue, and use CAGE only from that, do you see more tissue-specific signals? In the very least, does one lose out on tissue-specific genes when this combined approach is taken to identify promoters?

* Supplementary Figure 12 refered to on page 19 is for zebrafish. It should probably be Supplementary Figure 19. That supplementary section contains the word "knockout". The paragraph in the main paper only talks about "with and without the initiator sequence". What exactly was done? The results are very descriptive in nature, can any significance be associated with them?

* Citation needed for "It is known that dinucleotide profiles of DNA sequences hold more information than mononucleotide profiles" on page 23.

* What is Ckj in equation 2?

A few places where language can be improved:

1. Page 2:"happens by assembling the pre-initiation complex " -> "happens by assembly of the pre-initiation complex "

2. Page 3: authors say "as is traditional, has a limitation" but go on to list more than one.

3. Page 4: "suits well" -> "is well-suited"

4. Page 21: "fatorisation"

**Have the authors made all data and (if applicable) computational code underlying the findings in their manuscript fully available?**

Reviewer #1: Yes

Reviewer #2: Yes

Reviewer #3: Yes

PLOS authors have the option to publish the peer review history of their article (what does this mean?). If published, this will include your full peer review and any attached files.

Reviewer #1: No

Reviewer #2: No

Reviewer #3: No
---

## [Decision Letter · Decision Letter 1]

5 Sep 2023

Dear Dr Nikumbh,

We are pleased to inform you that your manuscript 'Identifying promoter sequence architectures via a chunking-based algorithm using non-negative matrix factorisation' has been provisionally accepted for publication in PLOS Computational Biology.

Please check potential discrepancies in the citations of some of the Supplementary figures in the revised manuscript:

P21, L402: According to the legend, Supp Fig 14 do not report data from human data, but from drosophila,

P21, L409: According to the legend, Supp Fig 29 do not report data from human data, but from Zebra fish,

P28, L664: Further similar discrepancies in citations of Supp Fig 12-14 and 17.

Before your manuscript can be formally accepted you will further need to complete some formatting changes, which you will receive in a follow up email. A member of our team will be in touch with a set of requests.

Best regards,

Denis Thieffry, PhD

Academic Editor

PLOS Computational Biology

William Noble

Section Editor

PLOS Computational Biology

Reviewer's Responses to Questions

**Comments to the Authors:**

Reviewer #3: Authors have addressed my concerns.

**Have the authors made all data and (if applicable) computational code underlying the findings in their manuscript fully available?**

Reviewer #3: None

PLOS authors have the option to publish the peer review history of their article (what does this mean?). If published, this will include your full peer review and any attached files.

Reviewer #3: No

---

## [Editor Report · Acceptance letter]

10 Nov 2023

PCOMPBIOL-D-23-00587R1 

Identifying promoter sequence architectures via a chunking-based algorithm using non-negative matrix factorisation

Dear Dr Nikumbh,

I am pleased to inform you that your manuscript has been formally accepted for publication in PLOS Computational Biology. Your manuscript is now with our production department and you will be notified of the publication date in due course.

With kind regards,

Judit Kozma
